# Uncertainty-Informed Meta Pseudo Labeling for Surrogate Modeling with Limited Labeled Data

**Xingyu Ren**[1][*]  **Pengwei Liu**[1][*]  **Pengkai Wang**[1][*]  **Guanyu Chen**[1]  **Qinxin Wu**[1]  **Dong Ni**[1][†]

[1] Zhejiang University, Hangzhou, China

{12332063,liupw,22032026,guanyu,22360247,dni}@zju.edu.cn

## Abstract

Deep neural networks, particularly neural operators, provide an efficient alternative to costly simulations in surrogate modeling. However, their performance is often constrained by the need for large-scale labeled datasets, which are costly and challenging to acquire in many scientific domains. Semi-supervised learning reduces label reliance by leveraging unlabeled data yet remains vulnerable to noisy pseudo-labels that mislead training and undermine robustness. To address these challenges, we propose a novel framework, Uncertainty-Informed Meta Pseudo Labeling (UMPL). The core mechanism is to refine pseudo-label quality through uncertainty-informed feedback signals. Specifically, the teacher model generates pseudo labels via epistemic uncertainty, while the student model learns from these labels and provides feedback based on aleatoric uncertainty. This interplay forms a meta-learning loop where enhanced generalization and improved pseudo-label quality reinforce each other, enabling the student model to achieve more stable uncertainty estimation and leading to more robust training. Notably, This framework is model-agnostic and can be seamlessly integrated into various neural architectures, facilitating effective exploitation of unlabeled data to enhance generalization in distribution shifts and out-of-distribution scenarios. Extensive evaluations of four models across seven tasks covering steady state and transient prediction problems demonstrate that UMPL consistently outperforms the best existing semi-supervised regression methods. When using only $10\%$ of the fully supervised training data, UMPL achieves a $14.18\%$ improvement, highlighting its strong effectiveness under limited supervision. Our codes are available at https://github.com/small-dumpling/UMPL.

## 1 Introduction

High-fidelity simulations are indispensable tools in scientific computing and engineering, enabling precise modeling of complex physical systems governed by partial differential equations (PDEs). They serve as foundational methodologies across diverse domains, including aerodynamics, chemical processing, and energy systems [64, 56, 59, 62, 39]. However, as the underlying systems grow increasingly nonlinear and coupled, the computational demands of high-fidelity simulations grow prohibitively high, limiting their feasibility for real-time deployment and large-scale iterative workflows [5, 32]. Recent advances in deep learning, particularly neural operator-based approaches, offer a promising alternative by introducing scalable, data-driven surrogate models capable of capturing complex spatiotemporal dynamics and learning solution mappings between function spaces [29, 3].

Nevertheless, the effectiveness of data-driven approaches is still heavily dependent on the quality and availability of training data. In cases where data are limited, these models tend to overfit

---

[*]These authors contributed equally.

[†]Corresponding author.

39th Conference on Neural Information Processing Systems (NeurIPS 2025).

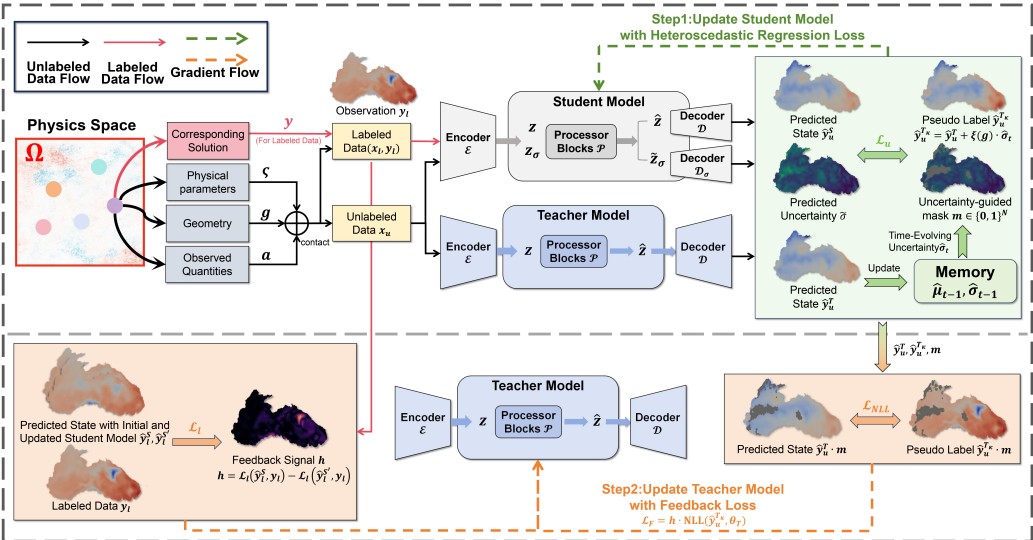

Figure 1: Overview of the UMPL pipeline. UMPL leverages both labeled and unlabeled data to enhance generalization under distribution shifts. The teacher generates pseudo labels via epistemic uncertainty, while the student learns from predicted states and their corresponding aleatoric uncertainties using heteroscedastic regression loss. The teacher is iteratively refined with feedback from the student's performance on labeled data, forming a closed-loop training process that progressively improves pseudo-label quality. Both the teacher and student can be versatile forecasting models.

and generalize poorly, restricting their use in situations where obtaining accurate simulations is expensive or unfeasible [13]. To mitigate these challenges, semi-supervised learning (SSL) has emerged as a promising solution by leveraging structural similarities between limited labeled data and abundant unlabeled data. Common SSL strategies often include pseudo-labeling [31, 19, 67], self-training [69, 8, 68], and contrastive learning [11, 4]. While SSL has shown promise in computer vision, its application to complex surrogate modeling remains underexplored [52]. The presence of strict physical constraints in scientific systems makes them particularly sensitive to the noise and inconsistencies introduced by conventional SSL methods [57], underscoring the urgent need for physically consistent semi-supervised frameworks.

In this paper, we propose Uncertainty-informed Meta Pseudo Labeling (UMPL), a novel semi-supervised learning framework explicitly designed for physical systems surrogate modeling. UMPL extends the meta pseudo-labeling paradigm by incorporating a teacher-student architecture informed by uncertainty, addressing key challenges such as limited labeled data and distributional shifts. The teacher model generates both predictions and corresponding epistemic uncertainty to construct uncertainty-aware pseudo labels. The student model is trained on these labels while estimating their heteroscedastic aleatoric inherent uncertainty. Throughout the training, the pseudo-label quality is assessed against a small labeled set, providing feedback that enables the teacher to refine its outputs iteratively. As a result, the student model can produce accurate predictions along with the relative uncertainty estimates. Extensive experiments across diverse physical-system datasets and forecasting architectures demonstrate that UMPL consistently outperforms existing semi-supervised learning baselines. An overview of the UMPL framework is presented in Figure 1. In summary, the contribution of our paper can be summarized as follows:

- **Problem Formulation.** To the best of our knowledge, this is the first attempt to integrate pseudo-labeling strategies with the modeling of high dimensional PDE systems on non-uniform meshes, aiming to address challenges for out-of-distribution generalization and distribution shifts arising from limited labeled data.

- **Role-Aware Dual-Uncertainty for Robust Generalization.** We design a teacher-student framework that assigns distinct uncertainty modeling strategies to each role, inspired by their functional differences and underlying physical data characteristics. The teacher models epistemic uncertainty to gauge confidence under distribution shifts, while the student models heteroscedastic aleatoric uncertainty to capture pseudo-label noise. This design improves generalization and uncertainty estimation on unseen data. Furthermore, the framework is flexible, allowing interchangeable predictive models and uncertainty estimators.

- **Uncertainty-Informed Feedback.** Building upon the dual-uncertainty modeling, we introduce an uncertainty-informed feedback mechanism: the student provides aleatoric uncertainty as a signal to guide the teacher in refining uncertain regions. This targeted feedback significantly improves the quality of pseudo labels across iterations, especially in high-uncertainty or underrepresented regimes.
- **Empirical Validation.** Extensive experiments conducted under various settings demonstrate the proposed UMPL framework's superior performance and robustness compared to existing semi-supervised learning methods.

## 2 Problem Setup

Consider the problem of learning solution operators for PDEs defined on a spatial domain $\Omega$. The objective is to learn a mapping $\mathcal{G} : \mathcal{A} \to \mathcal{S}$, where $\mathcal{A}$ and $\mathcal{S}$ denote the input and solution function spaces over $\Omega$. Each input $x = (g, a, \varsigma)$ consists of geometry $g \in \mathbb{R}^{N \times C_g}$ and associated physical quantities $a \in \mathbb{R}^{N \times C_a}$ and physical parameters $\varsigma$ observed on $g$; the corresponding solution is $y \in \mathbb{R}^{N \times C_s}$, where $N$ is the number of discretized spatial points. We approximate $\mathcal{G}$ using a neural network $\mathcal{G}_\theta$, which comprises three components: (1) a state encoder $\mathcal{E}$ that maps input features to latent embeddings $z \in \mathbb{R}^{N \times d_z}$, (2) a latent evolution module $\mathcal{P}$ that transforms $z$ to predicted latent states $\hat{z}$, and (3) a decoder $\mathcal{D}$ that maps $\hat{z}$ to the output solution $\hat{y}$. For transient systems, the prediction proceeds autoregressively over the time steps. Our goal is to closely approximate the underlying mapping $\mathcal{G}$ with the parameterized neural network. For training and evaluation, the supervision and evaluation loss is defined using the relative $L_2$ error:

$$\mathcal{L}_l(y, \hat{y}) = \frac{\| \boldsymbol{y} - \boldsymbol{\hat{y}} \|_2}{\| \boldsymbol{y} + \epsilon \|_2}, \tag{1}$$

where $y$ and $\hat{y}$ denote the prediction and target data, respectively. $\epsilon$ is a small constant introduced to ensure numerical stability. In practice, we consider the training data consists of a small labeled set $D_l = \{(x_l^i, y_l^i)\}_{i=1}^{n_l}$ and a larger unlabeled set $D_u = \{x_u^i\}_{i=1}^{n_u}$, with $n_u \gg n_l$. This setup reflects real-world constraints in scientific computing, where labeled data obtained from simulations is expensive, while unlabeled inputs are more readily available. The learning objective is to approximate the underlying solution operator $\mathcal{G}$ by utilizing labeled and unlabeled data.

## 3 Methodology

### 3.1 Asymmetric Uncertainty Estimation in Teacher and Student Models

To capture different sources of uncertainty, our framework adopts a dual-model design: the teacher estimates epistemic uncertainty via Monte Carlo (MC) Dropout, while the student models aleatoric uncertainty through direct variance prediction. While MC Dropout and ensemble methods approximate Bayesian inference, they incur high computational costs [36, 53, 66, 2]. In contrast, direct prediction methods are efficient but fail to capture model uncertainty, often yielding overconfident predictions on out-of-distribution data [45]. To balance these trade-offs, we apply MC Dropout in the teacher to quantify epistemic uncertainty, particularly relevant in simulation-based datasets with reliable labels but sparse input coverage. At the same time, the student learns from the teacher's pseudo labels and estimates aleatoric uncertainty to refine its predictions. This mutual guidance improves pseudo-label quality and promotes more robust learning.

**EMC Dropout for Teacher Model.** To further improve efficiency, we introduce Exponential Moving Average-enhanced MC Dropout (EMC Dropout), which incorporates exponential moving averages (EMA) into the estimation process. Inspired by Sequential MC Dropout [7], EMC Dropout incrementally updates the predictive mean and variance using a temporal sliding window, thereby preserving the Bayesian properties of dropout while avoiding repeated stochastic forward passes.

During the training progress, we set $\hat{\mu}_t$ as the mean value of prediction at time $t$, and variance $\hat{\sigma}_t^2$. At the initial time ($t = 0$), we perform a complete MC estimation using $M$ stochastic passes to obtain the initial predictive mean $\hat{\mu}_0$ and variance $\hat{\sigma}_0^2$. When get the prediction $\hat{y}_t$ at time $t$, we update the mean value $\hat{\mu}_{t+1}$ and variance $\hat{\sigma}_{t+1}^2$:

$$\begin{aligned} \hat{\mu}_{t+1} &= \alpha \hat{y}_t + (1 - \alpha)\hat{\mu}_t, \\ \hat{\sigma}_{t+1}^2 &= \alpha(\hat{y}_t - \hat{\mu}_{t+1})^2 + (1 - \alpha)\hat{\sigma}_t^2, \end{aligned} \tag{2}$$

where $\alpha \in (0, 1)$ serves as a gating factor that controls the balance between the influence of the most recent observation and the accumulated historical estimates. A higher value of $\alpha$ places more emphasis on recent data, while a lower value favors historical stability.

From a Bayesian perspective, dropout-based methods estimate the posterior predictive distribution by injecting stochasticity into the network's parameters, typically requiring multiple forward passes per input. EMC Dropout incorporates the EMA mechanism to reduce this computational overhead and track the predictive mean and variance over time. Rather than repeatedly sampling the stochastic model output, EMC Dropout incrementally updates running estimates during training. This strategy preserves the epistemic uncertainty captured by dropout while offering a more efficient approximation of Bayesian marginalization. Further analytical details are provided in Appendix B.1.

**Theorem 3.1. Asymptotic consistency of EMC Dropout.** Let $f_t(x)$ denote the prediction of a neural network with dropout applied at time step $t$, and $\hat{\mu}_t$, $\hat{\sigma}_t^2$ be the predictive mean and variance maintained over time with the gating factor $\alpha$. Suppose that $f_t(x) \sim p(y|x, w_t)$, where $w_t \sim q(w)$ represents a stochastic sample from the dropout-induced approximate posterior. Then, under stationarity and ergodicity assumptions on the sequence $\{f_t(x)\}_{t=1}^{\infty}$, the EMC estimates satisfy:

$$\hat{\mu}_t \to \mathbb{E}_{q(w)}[f(x; w)], \quad \hat{\sigma}_t^2 \to \mathbb{V}_{q(w)}[f(x; w)] \quad \text{as } t \to \infty. \tag{3}$$

Assume $\mu$ and $\sigma$ are the expectation estimation and variance of MC Dropout, we have:

$$\limsup_{t \to \infty} \mathbb{E}[(\hat{\mu}_t - \mu)^2] \leq \frac{\alpha}{2 - \alpha} \cdot \mathbb{E}_{q(w)}\left[(f_t(x; w) - \mu)^2\right],$$
$$\limsup_{t \to \infty} \mathbb{E}\left[(\hat{\sigma}_t^2 - \sigma^2)^2\right] \leq \frac{\alpha}{2 - \alpha} \cdot \text{Var}_{q(w)}[(f_t(x; w) - \mu)^2]. \tag{4}$$

The proof of Theorem 3.1 is provided in Appendix B.2. It establishes that, under a sufficiently mixing sampling process, the EMA estimates of the predictive mean and variance converge almost surely to the Monte Carlo estimates of the Bayesian predictive distribution induced by dropout.

**Model-Based Uncertainty Estimation for Student Model.** To account for the noise and uncertainty inherent in pseudo labels provided by the teacher model, we train a student model to predict both the target value and an input-dependent variance. Inspired by uncertainty quantification techniques in PDEs forward problems[63], we introduce an additional uncertainty decoder to estimate this variance. Given an input $x$, the model uses an auxiliary decoder $\mathcal{D}_\sigma$ to produce the corresponding predictive uncertainty $\tilde{\sigma} \in \mathbb{R}^{N \times C_s}$ from the latent state $\tilde{z}_\sigma$. Then, we optimize the student model using a $L_2$ relative error based heteroscedastic regression loss based on the pseudo labels:

$$\mathcal{L}_u(y, [\hat{y}, \tilde{\sigma}]) = \frac{1}{N_u} \sum_{i=1}^{N_u} \left( \frac{\|y_i - \hat{y}_i\|_2}{\tilde{\sigma}_i^2 \cdot \|y_i + \epsilon\|_2} + \log \tilde{\sigma}_i^2 \right), \tag{5}$$

where $\hat{y}$ denotes the prediction of the target label $y$. The loss function down-weights errors for samples with higher predicted aleatoric uncertainty, reflecting their lower label reliability [25]. Its formal derivation is based on maximum likelihood estimation under the assumption of heteroscedastic observation errors [24], enabling more effective student optimization and providing informative feedback for the teacher.

## 3.2 Uncertainty-Informed Feedback for Teacher Model Update

We adopt an uncertainty-informed strategy that replaces standard meta-gradient computation to enable efficient teacher updates without incurring the high cost of per-sample gradient tracing. While the standard MPL [47] method relies on categorical pseudo labels and is limited to classification. it does not generalize well to regression tasks with continuous outputs. Alternatives like Noisy TS [57] use noisy teacher predictions and scalar feedback but require manually tuned noise and apply uniform feedback across the input space, leading to spatial inconsistency and limited adaptability. In contrast, our method introduces structured, point-wise feedback informed by model uncertainty, enabling coherent and efficient teacher updates in high-dimensional regression settings. Details of the feedback loss during training are illustrated in Figure 2. The feedback loss produced by our method on unlabeled data more closely approximates the actual loss, indicating more reliable guidance than Noisy TS [57], which is a current meta pseudo-labeling approach designed to improve model performance in ODE-based dynamical system modeling.

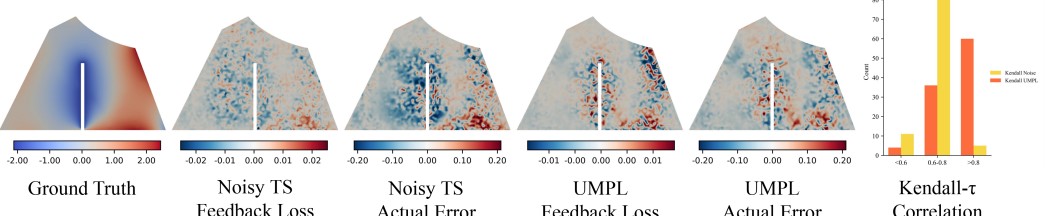

Figure 2: Figure 2. Comparison of feedback loss and true pseudo-label error during training for UMPL and Noisy TS [57] using AeroGTO [38] as the base forecaster. The rightmost subplot illustrates the $Kendall\text{-}\tau$ correlation between feedback loss and true pseudo-label error across 100 unlabeled cases, showing that UMPL's aleatoric uncertainty-guided feedback more reliably reflects pseudo-label error.

**Uncertainty-Informed Pseudo Label Generation.** We design the teacher model to integrate predictive uncertainty into pseudo label generation, thereby enhancing the reliability and informativeness of the supervisory signals. For simplicity, the following definitions and derivations are provided for a single prediction step, without incorporating multi-step autoregression. Given the teacher's variance $\hat{\sigma}_t^2$ at time t, we can get the pseudo labels:

$$\hat{y}_u^{T_\kappa} = \hat{y}_u^T(\theta_T) + \xi(g) \cdot \hat{\sigma}_t, \tag{6}$$

where $\xi(g)$ denotes a random field obtained through Gaussian smoothing over the geometry $g$, which encourages continuity in the generated pseudo labels. Specifically, the geometric mesh is composed of points and edges, and the smoothing is performed based on the local connectivity defined by the mesh structure:

$$\xi(g_i) = \text{sign}\left(\sum_{j \in \mathcal{U}(i)} \exp\left(-\frac{\|g_i - g_j\|^2}{2l^2}\right) \cdot r_i\right), \quad r_i \sim \mathcal{N}(0, 1), \tag{7}$$

where $\mathcal{U}(i)$ is the geometric neighborhood of the point $i$, defined by edge connectivity for meshes or k-nearest neighbors for point clouds. $g_i, g_j$ are spatial coordinates, $l$ controls the smoothing scale, and $r_j \sim \mathcal{N}(0, 1)$ is sampled noise. The sign $\varepsilon_i \in \{-1, +1\}$ encodes smoothed perturbations with spatial coherence.

**Derivation of the Feedback Gradient.**

To optimize the teacher model based on the student's feedback, the objective is to minimize the following loss:

$$\min \mathcal{L}_l(\theta_S^{PL}(\theta_T)),$$
$$\text{where } \theta_S^{PL}(\theta_T) = \underset{\theta_S}{\text{argmin}} \ \mathcal{L}_u(\theta_T, \theta_S), \tag{8}$$

where $\mathcal{L}_l$ and $\mathcal{L}_u$ denote the losses on labeled data and pseudo-labeled data, respectively. The argmin operation in the above equation poses a challenge for gradient-based optimization, as it requires the parameter $\theta_S$ to attain its optimal value before the subsequent step can proceed. This inherently breaks the end-to-end differentiability of the training pipeline. To overcome this issue, we employ a one-step approximation that allows for tractable and efficient optimization.

$$\theta_S^{PL}(\theta_T) \approx \theta_S - \eta_S \cdot \nabla_{\theta_S} \mathcal{L}_u(\theta_T, \theta_S), \tag{9}$$

Here, the optimization goal becomes:

$$\min_{\theta_T} \mathcal{L}_l(\theta_S - \eta_S \cdot \nabla_{\theta_S} \mathcal{L}_u(\theta_T, \theta_S)). \tag{10}$$

In principle, if the gradient of the objective concerning $\theta_T$ is computable, end-to-end optimization via gradient descent becomes feasible. Denoting Eq. 10 as $\mathcal{L}_F$, the gradient can be derived using the

REINFORCE rule [60] as follows:

$$
\begin{aligned}
\frac{\partial \mathcal{L}_F}{\partial \theta_T} =& \frac{\partial}{\partial \theta_T} \mathcal{L}_l \left( y_l, S \left( x_l; \mathbb{E}_{\hat{y}_u^{T_\kappa} \sim T(x_u; \theta_T)} \left[ \theta_S - \eta_S \nabla_{\theta_S} \mathcal{L}_u(\hat{y}_u^{T_\kappa}, S(x_u; \theta_S)) \right] \right) \right) \\
=& \eta_S \cdot \frac{\partial \mathcal{L}_l(y_l, S(x_l; \theta_S'))}{\partial \theta_S} \cdot \left( \left. \frac{\partial \mathcal{L}_u(\hat{y}_u^{T_\kappa}, S(x_u; \theta_S))}{\partial \theta_S} \right|_{\theta_S = \theta_S} \right)^\top \cdot \frac{\partial \log p(\hat{y}_u^{T_\kappa} | x_u; \theta_T)}{\partial \theta_T} \quad (11) \\
=& \eta_S \cdot \left( \nabla_{\theta_S'} \mathcal{L}_l(y_l, S(x_l; \theta_S'))^\top \cdot \nabla_{\theta_S} \mathcal{L}_u(\hat{y}_u^{T_\kappa}, S(x_u; \theta_S)) \right) \cdot \nabla_{\theta_T} p(\hat{y}_u^{T_\kappa} | x_u; \theta_T) \\
\approx& h \cdot \mathrm{NLL}(\hat{y}_u^{T_\kappa}; \theta_T),
\end{aligned}
$$

where $h = \mathcal{L}_l(y_l, S(x_l; \theta_S)) - \mathcal{L}_l(y_l, S(x_l; \theta_S'))$ denotes the change in student performance after training on a pseudo label and $NLL := -\log p(y|x; \theta)$ is a scaled negative log-likelihood (NLL) loss. If $h \geq 0$, minimizing $L_F$ increases the likelihood of similar pseudo labels in future updates. otherwise, it suppresses them. More details about the feedback loss can be found in Appendices B.3 and B.4. Combined with the supervised loss $\mathcal{L}_l^T := \mathcal{L}_l(y_l, T(x_l; \theta_T))$, the teacher's update is performed as follows:

$$
\theta_T' = \theta_T - \eta_T \cdot \nabla_{\theta_T} \cdot (\mathcal{L}_l^T + \mathcal{L}_F). \quad (12)
$$

**Theorem 3.2. Generalization error bound of the student.**

Let $\mathcal{F}$ denote the hypothesis class of neural surrogate models with fixed architecture and bounded parameters. With the training set consisting of $n_l$ labeled and $n_u$ pseudo-labeled samples, for any $\delta \in (0, 1)$, with probability at least $1 - \delta$, the following inequality holds for the student model $S(x_u'; \theta_S) \in \mathcal{F}$ where $x_u'$ denotes an unseen input drawn from the underlying data distribution:

$$
\mathbb{E}_{(x, y_u)} \left[ \mathcal{L}_l(y_u, S(x_u; \theta_S)) \right] \leq \mathcal{L}_{\mathrm{semi}}(S(\theta_S)) + 2L \cdot \hat{\mathfrak{R}}_{n_l + n_u}(\mathcal{F}) + c \cdot \sqrt{\frac{\log(1/\delta)}{2(n_l + n_u)}} + \lambda_p \cdot \varepsilon_{\mathrm{pseudo}}^{\mathrm{rel}},
$$
(13)

where $\mathcal{L}_{semi}$ denotes the empirical loss computed over both labeled and pseudo-labeled data. $\hat{\mathfrak{R}}_{n_l + n_u}(\mathcal{F})$ is the empirical Rademacher complexity of the hypothesis class $\mathcal{F}$; $L$ represents the Lipschitz constant of the loss function. and $c$ is a universal constant. The coefficient $\lambda_p \in [0, 1]$ controls the relative contribution of the pseudo-labeled samples. $\varepsilon_{\mathrm{pseudo}}^{\mathrm{rel}} := \mathbb{E}_{x \sim D_u} \left[ \mathcal{L}_u(\hat{y}_u^{T_\kappa}, S(x_u; \theta_S)) \right]$ denotes the expected error between the pseudo-labels and the student's predictions on unlabeled inputs. The proof can be seen in Appendix B.2.

This bound implies that the student model can achieve improved generalization performance when the reduction in model complexity exceeds the additional error introduced by the pseudo-labels.

### 3.3 Training Workflow of the UMPL Framework

We propose a unified training framework that optimizes teacher and student models through mutual feedback. To improve robustness against noisy pseudo-labels, we incorporate Progressive Uncertainty Filtering (PUF), which leverages EMC Dropout to estimate predictive uncertainty. Based on this, a binary mask $m \in \{0, 1\}^N$ is constructed by selecting the top-k% most uncertain predictions. The corresponding masked regions ($m_i = 1$) are excluded from the student's loss computation and the feedback used to update the teacher. As training progresses and the reliability of pseudo labels improves, the masking ratio is gradually decreased, enabling the student to learn from a broader range of samples and provide more informative feedback to the teacher.

After initialization, the training process follows three main steps: (1) **Pseudo-label generation:** The teacher model computes the prediction and variance with Eq.2, generating pseudo-labels ( Eq.6). (2) **Student training on pseudo labels:** Train the student with an unlabeled loss defined on pseudo labels with Eq.5, and estimate the improvement $h$ as the student feedback. (3) **Teacher and student's update:** The teacher is updated using both the supervised loss and the feedback loss (Eq. 12), while the student is updated exclusively on labeled data. This strategy, referred to as Supervised Anchoring (SA), ensures the student remains grounded in reliable supervision, preventing error accumulation during self-training, especially under domain shifts where pseudo-labels may be unreliable. The design of the algorithm ensures continued exploration of unseen data space and mitigates the risk of model collapse. A detailed description of the algorithm is presented in Algorithm 1. More details about training can be found in Appendix C.

**Algorithm 1** UMPL Algorithm

---

**Input:** Labeled data: $D_l = \{(x_l^i, y_l^i)\}_{i=1}^{n_l}$, unlabelled data: $D_u = \{x_u^i\}_{i=1}^{n_u}$, teacher and student neural operator: $T(\cdot; \theta_T), S(\cdot; \theta_S)$, epoch $N_{epoch}$
**Output:** Trained student: $S(\cdot; \theta_S^*)$
**for** $t = 0, 1, 2, \cdots, N_{epoch} - 1$ **do**

1. Sample a batch of labeled data $\{(x_l^t, y_l^t)\}$ and an equal-sized batch of unlabeled data $x_u^t$.

2. Generate pseudo rollouts from the teacher and record the old student's performance on the labeled data:
$$\hat{y}_u^{T_\kappa} = \hat{y}_u^T(\theta_T) + \xi(g) \cdot \hat{\sigma}_t,$$
$$\mathcal{L}_l^{S^t} = \mathcal{L}_l(y_l^t, S(x_l^t, \theta_S^t))$$

3. Update the student with pseudo loss and record the new student's performance on the same labeled data:
$$\theta_S^{t+1} = \theta_S^t - \eta_S \cdot \nabla_{\theta_S^t} \mathcal{L}_S^u,$$
$$\mathcal{L}_l^{S^{t+1}} = \mathcal{L}_l(y_l^t, S(x_l^t, \theta_S^{t+1}))$$

4. Calculate the feedback loss:
$$\mathcal{L}_F = (\mathcal{L}_l^{S^t} - \mathcal{L}_l^{S^{t+1}}) \cdot \text{NLL}(y_u^{T_\kappa}; \theta_T^t)$$

5. Update the teacher with its labeled loss and the feedback loss and update the student with its labeled loss:
$$\theta_T^{t+1} = \theta_T^t - \eta_T \cdot \nabla_{\theta_T^t}(\mathcal{L}_l^T + \mathcal{L}_F),$$
$$\theta_S^{t+1} = \theta_S^{t+1} - \eta_S \cdot \nabla_{\theta_S^{t+1}} \mathcal{L}_l^{S^{t+1}}$$

**end for**
**return** $S(\cdot; \theta_S^{N_{epoch}})$

---

## 4 Experiment

### 4.1 Experimental Settings

**Benchmarks.** We study datasets from three domains, as summarized below. **Partial Differential Equations.** We consider four PDE benchmarks: (i) Lid-driven cavity flow [16], where a constant top boundary velocity induces circulation. We explore steady and transient settings, using Reynolds number($Re$) and lid velocity (v) as orthogonal parameters for out-of-distribution generalization and distribution shifts evaluation. (ii) Darcy flow [9], a classical model describing fluid transport in porous media. (iii) NSM2D [37], the 2D Navier-Stokes equations augmented with a magnetic field component. (iv) Plasma ICP, where low-resolution simulations with incomplete chemical reaction mechanisms are used for pretraining, followed by fine-tuning on limited high-resolution data to assess robustness under distribution shifts. **Real-world Data.** We employ the Black Sea dataset [10], which contains real-world ocean temperature measurements over time, to evaluate temporal out-of-distribution generalization. **Large-scale 3D Simulation.** We use the Ahmed dataset [34], where the input is the 3D geometry of a vehicle, and the output is its corresponding aerodynamic drag coefficient. This benchmark evaluates the model's ability to generalize aerodynamic predictions across diverse vehicle shapes.

**Baselines.** We evaluate our method on four representative surrogate modeling architectures, including two point-based models (DeepONet [41] and Transolver [61]) and two mesh-based models (MeshGraphNet(MGN) [46] and AeroGTO [38]), to assess its performance across different data representations. We compare UMPL against four baseline pseudo-labeling strategies (more details see Appendix D): **Supervision Only**. The model is trained exclusively on labeled data. **Pseudo Label (PL)** [31]. The teacher remains fixed, and the student is trained on both labeled data and pseudo labels generated by the teacher. **Mean Teacher** [54]. A widely used and stable approach in which the teacher is updated via EMA of the student's parameters. **Noisy TS** [57]. A meta pseudo-labeling method where the teacher receives feedback from the student and generates pseudo labels with added

Table 1: We compare the performance of our method against four baselines across four different models under a labeled data ratio of $r = 10\%$ for all tasks. **Cyan** , **Yellow** , and **Green** indicate the best, second-best, and worst $L_2$ loss values, respectively. \ indicates non-convergence.

| Model | Method | Darcy Flow | Ahmed | NSM2d | Plasma ICP | Stationary Lid | | Black Sea | |
|---|---|---|---|---|---|---|---|---|---|
| | | | | | | w/o OOD | w/ OOD | In-T | Out-T |
| DeepONet[41] | Supervision Only | 7.98e-2 | \ | 3.07e-1 | \ | 6.07e-2 | 1.69e-1 | 1.45e-1 | 2.41e-1 |
| | Pseudo Label[31] | 8.37e-2 | \ | 2.82e-1 | \ | 5.65e-2 | 1.73e-1 | 1.51e-1 | 2.35e-1 |
| | Mean Teacher[54] | 8.36e-2 | \ | 2.77e-1 | \ | 5.57e-2 | 1.77e-1 | 1.46e-1 | 2.22e-1 |
| | Noisy TS[57] | 7.83e-2 | \ | 2.57e-1 | \ | 5.38e-2 | 1.42e-1 | 1.24e-1 | 1.67e-1 |
| | UMPL(ours) | 6.94e-2 | \ | 2.23e-1 | \ | 4.57e-2 | 1.08e-1 | 9.61e-2 | 1.43e-1 |
| | **PROMOTION** | **11.36%** | \ | **13.22%** | \ | **15.05%** | **23.94%** | **22.50%** | **14.37%** |
| MGN[46] | Supervision Only | 8.06e-2 | 9.11e-2 | 5.62e-1 | \ | 3.82e-2 | 1.02e-1 | 9.84e-2 | 1.13e-1 |
| | Pseudo Label[31] | 7.33e-2 | 8.76e-2 | 4.95e-1 | \ | 2.22e-2 | 1.13e-1 | 9.97e-2 | 1.08e-1 |
| | Mean Teacher[54] | 7.33e-2 | 8.32e-2 | 4.41e-1 | \ | 1.97e-2 | 1.07e-1 | 9.83e-2 | 1.07e-1 |
| | Noisy TS[57] | 6.49e-2 | 6.92e-2 | 3.85e-1 | \ | 2.20e-2 | 9.48e-2 | 8.51e-2 | 9.67e-2 |
| | UMPL(ours) | 5.74e-2 | 6.34e-2 | 3.51e-1 | \ | 1.64e-2 | 7.91e-2 | 8.19e-2 | 8.99e-2 |
| | **PROMOTION** | **11.55%** | **8.38%** | **8.83%** | \ | **16.75%** | **16.56%** | **3.76%** | **7.03%** |
| Transolver[61] | Supervision Only | 6.14e-2 | 7.69e-2 | 2.71e-1 | 1.90e-1 | 3.69e-2 | 9.55e-2 | 9.34e-2 | 1.12e-1 |
| | Pseudo Label[31] | 5.24e-2 | 7.30e-2 | 2.35e-1 | 1.85e-1 | 1.79e-2 | 1.08e-1 | 9.16e-2 | 9.79e-2 |
| | Mean Teacher[54] | 5.34e-2 | 7.21e-2 | 2.21e-1 | 1.72e-1 | 2.19e-2 | 1.31e-1 | 9.26e-2 | 9.68e-2 |
| | Noisy TS[57] | 4.66e-2 | 5.89e-2 | 1.79e-1 | 1.56e-1 | 1.64e-2 | 9.53e-2 | 8.81e-2 | 9.17e-2 |
| | UMPL(ours) | 4.37e-2 | 5.17e-2 | 1.51e-1 | 1.41e-1 | 1.24e-2 | 7.32e-2 | 7.74e-2 | 8.53e-2 |
| | **PROMOTION** | **6.22%** | **12.22%** | **15.64%** | **9.61%** | **24.39%** | **23.18%** | **12.14%** | **6.97%** |
| AeroGTO[38] | Supervision Only | 3.82e-2 | 6.88e-2 | 1.72e-1 | 9.30e-2 | 3.51e-2 | 9.44e-2 | 8.95e-2 | 1.10e-1 |
| | Pseudo Label[31] | 3.17e-2 | 5.85e-2 | 1.51e-1 | 7.20e-2 | 8.75e-3 | 1.17e-1 | 8.24e-2 | 1.03e-1 |
| | Mean Teacher[54] | 3.03e-2 | 5.86e-2 | 1.45e-1 | 6.97e-2 | 2.17e-2 | 1.24e-1 | 9.07e-2 | 1.02e-1 |
| | Noisy TS[57] | 2.85e-2 | 4.71e-2 | 1.39e-1 | 6.24e-2 | 1.09e-2 | 7.98e-2 | 7.42e-2 | 9.38e-2 |
| | UMPL(ours) | 2.56e-2 | 3.63e-2 | 1.21e-1 | 5.25e-2 | 6.91e-3 | 6.33e-2 | 6.66e-2 | 8.12e-2 |
| | **PROMOTION** | **10.17%** | **22.92%** | **12.94%** | **15.86%** | **21.02%** | **20.67%** | **10.78%** | **13.43%** |

random noise. As data augmentation and noise injection may harm model accuracy [23, 57], we do not employ any augmentation techniques during training.

**Tasks.** We evaluate UMPL under three regimes with a unified protocol. **Limited-label setting.** On Ahmed, Darcy Flow, and NSM2D, we vary the labeled ratio $r \in \{10\%, 20\%, 30\%\}$. For each $r$, a stratified labeled subset is fixed and the remaining training samples are treated as unlabeled. The test split is held fixed across $r$ to ensure comparability. **Out-of-distribution generalization.** On the stationary Lid dataset, training and test sets are drawn from disjoint parameter regions to probe out-of-distribution parameter space. On the Black Sea dataset, models are trained on earlier, contiguous time windows and evaluated on non-overlapping future windows to probe out-of-distribution temporal space. **Distribution shifts.** We evaluate domain adaptation on the time-dependent lid dataset via cross-Reynolds-number transfer ($Re = 500 \longleftrightarrow 5000$) and on the Plasma ICP dataset by adapting from low-fidelity simulations to high-fidelity measurements. Results are averaged over five independent runs with different random seeds. Details regarding the datasets, models, and experimental settings can be found in Appendix E.

Table 2: Evaluation on four models across datasets under label ratios $r = 10\%, 20\%, 30\%$.

| Model | Black Sea | | | Time-dependent Lid2d | | |
|---|---|---|---|---|---|---|
| | 10% | 20% | 30% | 10% | 20% | 30% |
| DeepONet | 1.93e-1 | 1.16e-1 | 9.95e-2 | 3.34e-1 | 2.39e-1 | 2.16e-1 |
| DeepONet+UMPL | 1.19e-1 | 8.33e-2 | 8.09e-2 | 1.90e-1 | 1.59e-1 | 1.38e-1 |
| MGN | 1.05e-1 | 9.28e-2 | 7.12e-2 | 3.00e-1 | 1.64e-1 | 8.97e-2 |
| MGN+UMPL | 8.99e-2 | 8.05e-2 | 5.56e-2 | 2.65e-1 | 9.60e-2 | 4.65e-2 |
| Transolver | 1.02e-1 | 7.78e-2 | 6.28e-2 | 2.15e-1 | 9.53e-2 | 6.27e-2 |
| Transolver+MPL | 8.94e-2 | 6.54e-2 | 5.06e-2 | 2.04e-1 | 7.85e-2 | 5.17e-2 |
| AeroGTO | 1.00e-1 | 7.27e-2 | 5.98e-2 | 7.24e-2 | 3.54e-2 | 1.93e-2 |
| AeroGTO+MPL | 7.69e-2 | 5.88e-2 | 4.57e-2 | 4.63e-2 | 1.85e-2 | 1.06e-2 |

Table 3: Ablation results of different combinations of *SA*, *PUF*, and *Feedback* modules.

| SA | PUF | Feedback | $L_2$ Error |
|---|---|---|---|
| × | × | × | 1.06e-1 |
| ✓ | × | × | 8.70e-1 |
| × | × | ✓ | 8.46e-2 |
| ✓ | × | ✓ | 6.72e-2 |
| × | ✓ | ✓ | 8.18e-2 |
| ✓ | ✓ | ✓ | **5.93e-2** |

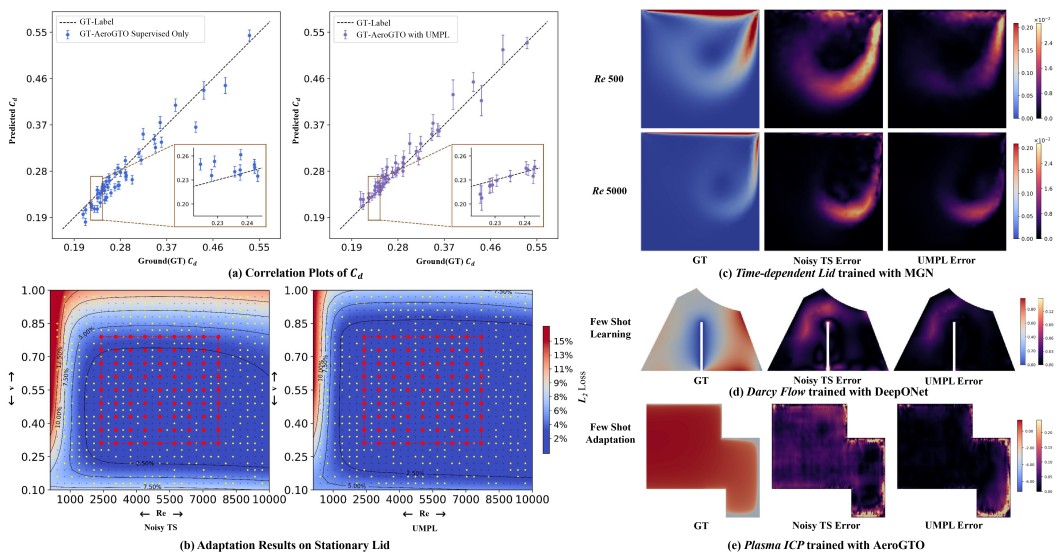

Figure 3: (a) Correlation between predicted drag coefficient $C_d$ and uncertainty from UMPL and the fully supervised model, compared to ground truth on the Ahmed test set (51 unseen geometries). (b) $L_2$ loss over the whole parameter domain on the Stationary Lid dataset. •, ∗, and ·, denote labeled, unlabeled, and test points, respectively. The red dashed boundary marks the in-distribution region. outside areas represent the out-of-distribution region. (c-e) Visualization results are presented across different datasets, with transient cases shown at the final time step.

## 4.2 Main Results

**Performance Comparison.** Table 1 presents the performance of UMPL across three representative tasks under limited-label conditions. On steady-state datasets (Darcy Flow, Ahmed Body) and the transient dataset (NSM2d), UMPL consistently outperforms all baselines, yielding an average improvement of 12.13% over the second-best approach. In out-of-distribution scenarios (Stationary Lid, Black Sea), UMPL achieves the lowest $L_2$ errors both within and beyond the training distribution, surpassing the second-best method by 20.19% and 11.37%, respectively. Likewise, distribution shift benchmarks (Time-dependent Lid2d and Plasma ICP) demonstrate superior performance. As shown in Table 2, UMPL consistently improves accuracy across varying label ratios, with particularly notable gains under the most challenging label distribution setting. These results collectively highlight the method's strong generalization capability and data efficiency.

**Temporal and Spatial Generalization.** UMPL consistently outperforms all baselines across diverse tasks, demonstrating strong generalization capabilities in both temporal and parametric domains. To further illustrate this, we present detailed visualizations of the Stationary Lid dataset. As shown in Figure 3(b), UMPL achieves lower $L_2$ errors than the previous best-performing method, Noisy TS, particularly in out-of-distribution regions. Moreover, UMPL maintains lower testing loss and better training stability compared to the PL method, which exhibits early-stage overfitting. The corresponding learning curves and detailed comparisons are provided in Appendix F.

**Visualization and Analysis.** Figure 3(a) illustrates the performance gains of UMPL over the Supervised Only baseline on the Ahmed Body dataset. Our method not only achieves more accurate predictions, but also produces better-calibrated uncertainty estimates. In contrast, the Supervised Only model yields overconfident and homogeneous uncertainty, failing to reflect the true variability and uncertainty in the data. Figure 3(b) presents the error distribution on the test set, showing that UMPL produces tighter and more favorable error profiles compared to existing methods. Figure 3(c-e) further compares UMPL with baselines across additional datasets, consistently demonstrating its advantage in both prediction accuracy and uncertainty calibration. Figure 4(a) shows performance on the Black Sea dataset, where UMPL achieves lower prediction error and provides well-structured uncertainty estimates. Figure 4(b) presents a temporal rollout on the NSM2d dataset, highlighting UMPL's ability to capture complex system dynamics with lower error and more robust uncertainty estimation compared to baseline approaches. Together, these results demonstrate that UMPL offers stronger generalization under both distribution shifts and out-of-distribution scenarios. Moreover, the

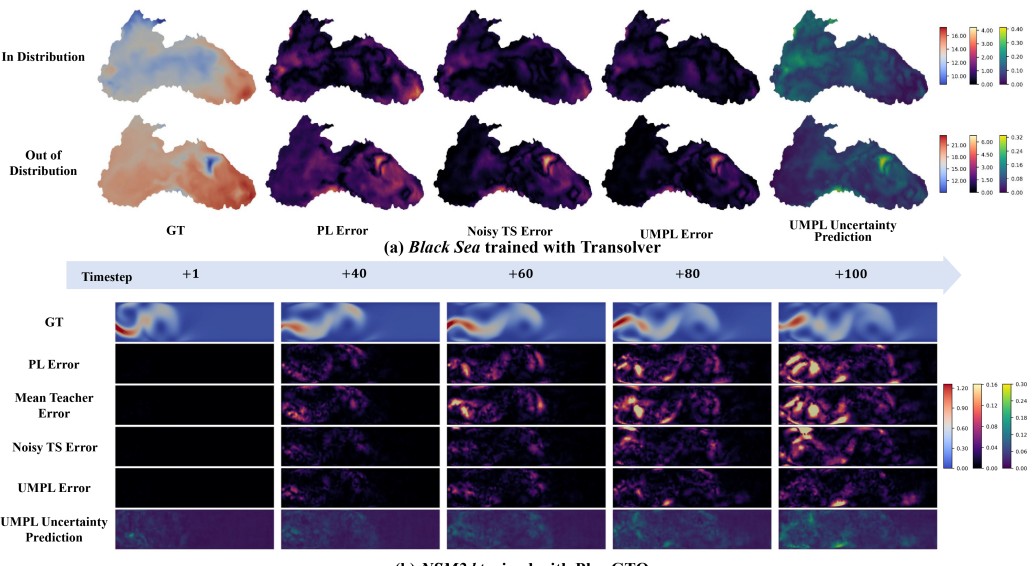

Figure 4: Visualization results across multiple methods and models are presented, where all temporal predictions are visualized at the final time step. The results show that the UMPL framework significantly reduces prediction error and provides more reliable uncertainty estimates.

student model trained under our uncertainty-informed meta pseudo-labeling framework consistently provides reliable and interpretable uncertainty estimates across diverse tasks.

## 4.3 Ablation Studies

We perform ablation studies on the Stationary Lid dataset to investigate the impact of each component within the UMPL framework. The experiments reveal that removing either the uncertainty-based pseudo-label filtering (PUF) or the supervised alignment (SA) module leads to a clear degradation in model performance, while eliminating both results in the most significant decline. Notably, using only the supervised component reduces the method to conventional pseudo-labeling and fails to match the performance of the full framework. These observations underscore the complementary roles of PUF and SA, demonstrating that both are essential for achieving robust and accurate predictions. Further analysis on hyperparameter sensitivity is included in Appendix F.5.

## 5 Discussion and limitations

**Limitations.** While the proposed method consistently improves performance, it still exhibits several limitations. It assumes that training and testing data are identically distributed (i.i.d.), causing performance to degrade under significant distribution shifts, and it struggles to adapt to highly dynamic and complex environments. Furthermore, the approach introduces computational and storage overhead due to the required EMC Dropout in the teacher model. Its loss function also necessitates aligned output shapes, which restricts its applicability to scenarios involving non-uniform or adaptive meshes. Future work should focus on improving computational efficiency and better leveraging meta pseudo-labels for more complex applications, like adaptive 3D flow field modeling.

**Conclusion.** This paper tackles the challenge of the surrogate model under out-of-distribution generation and distribution shift scenarios with limited labeled data. We propose UMPL, a semi-supervised framework that leverages uncertainty to refine pseudo-labeling: the teacher estimates epistemic uncertainty while the student captures aleatoric uncertainty. These signals guide learning from confident pseudo labels and enable feedback via labeled data, allowing the teacher to reinforce reliable predictions and suppress errors. Experiments show that UMPL reduces error in uncertain regions and enhances generalization. Future work includes extending UMPL to tasks like 3D flow prediction.

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

Table 4: Table of notations.

| Notation | Meaning |
|---|---|
| **Problem Formulation** | |
| $\Omega$ | spatial domain for PDEs |
| $\mathcal{G}$ | the mapping between input space and solution space |
| $\mathcal{A}$ | the input function space |
| $\mathcal{S}$ | the solution function space |
| $x = (g, a, \varsigma)$ | the input function for neural networks |
| $g \in \mathbb{R}^{N \times C_g}$ | the input geometry spatial coordinates |
| $a \in \mathbb{R}^{N \times C_a}$ | the physical quantities on the input geometry |
| $\varsigma$ | the coefficients, such as viscosity |
| $y \in \mathbb{R}^{N \times C_s}$ | the corresponding solution |
| $\mathcal{E}, \mathcal{P}, \mathcal{D}$ | the encoder, preocessor and decoder for neural operators |
| $z \in \mathbb{R}^{N \times d_z}$ | the latent states |
| $\hat{y}, \hat{z}$ | predicted latent states and corresponding output solution |
| $D_l = \{(x_l^i, y_l^i)\}_{i=1}^{n_l}$ | Labelled data |
| $D_u = \{x_u^i\}_{i=1}^{n_u}$ | unlabelled data |
| $\mathcal{U}_i$ | the geometric neighborhood of the point $i$ |
| **Notations in UMPL Training** | |
| $\mathcal{L}_l(y, \hat{y})$ | the relative $L_2$ error for labelled data |
| $\mathcal{L}_u(y, [\hat{y}, \tilde{\sigma}])$ | the heteroscedastic regression loss for pseudo labels |
| $\mathcal{L}_F$ | the feedback loss for the teacher |
| $\hat{\mu}_t, \hat{\sigma}_t$ | the EMC estimation at time $t$ |
| $\mu_t, \sigma_t$ | the expectation estimation and variance of MC Dropout at time $t$ |
| $\tilde{\sigma}$ | the uncertainty estimation from the model |
| $\xi(g)$ | a random field obtained through Gaussian smoothing over $g$ |
| $f_t(x)$ | the prediction of the model at time $t$ |
| $w_t \sim q(w)$ | a sample from the dropout-induced posterior at time $t$ |
| $\theta_S, \theta_T$ | the parameters for the teacher and student |
| $-\log p(y|x; \theta)$ | the scaled negative log-likelihood (NLL) loss |
| $h$ | improvement for student from pseudo-label training |
| $\alpha$ | the gating factor of forgetting in EMC Dropout |
| $r_i$ | sampled Gaussian noise at point $i$ |
| $l$ | the smoothing scale factor |
| $m$ | the binary mask for the progressive uncertainty filtering |
| $k$ | the masking ratio |
| **Notations in Theoretical Analysis** | |
| $\hat{\mathfrak{R}}_{n_l+n_u}(\mathcal{F})$ | the empirical Rademacher complexity of the hypothesis class $\mathcal{F}$ |
| $\lambda_p$ | the weight of the pseudo-labeled component |
| $\lambda_d$ | the weight decay coefficient |
| $\tau$ | expected model precision |
| $p$ | dropout rate |
| $\varepsilon_{pseudo}^{rel}$ | the expected relative error of the pseudo-labels |
| **Methodology** | |
| PL | pseudo labels |
| UMPL | uncertainty-informed meta pseudo labels |
| PUF | progressive uncertainty filtering |
| SA | supervised anchoring |
| EMA | Exponential Moving Average |
| EMC | MC Dropout with EMA |

In this appendix, we complement the main paper with additional theory, implementation details, and extended experiments. **Appendix** A surveys related work and situates our contributions. **Appendix** B provides supplementary theory, including a Bayesian interpretation of MC Dropout, proofs of Theorems 3.1 and 3.2, gradient derivations for the teacher update, and an analysis of feedback-gradient bottlenecks under a deterministic teacher. **Appendix** C reports training configurations and optimization details, while **Appendix** D describes the compared approaches. **Appendix** E details datasets, model architectures, and experimental settings. **Appendix** F presents extended evaluations—additional comparisons, computational cost analysis, robustness assessments, and ablations—along with sensitivity studies on uncertainty perturbations, PUF thresholds, feedback strength, and other hyperparameters to elucidate model behavior. Finally, **Appendix** G discusses broader impact.

## A    Related works

**Neural Operator for surrogate modeling.** Neural operator-based methods, such as Deep Operator Networks (DeepONet) [41], Fourier Neural Operators (FNO) [33], and MeshGraphNet (MGN) [46], have shown promise in learning mappings between infinite-dimensional function spaces, providing end-to-end frameworks for modeling parametric partial differential equations (PDEs). Transformer-based architectures, including Graph Neural Operator Transformers (GNOT) [17], Transolver [61], and AeroGTO [38], further push the boundaries of dynamic modeling by effectively handling large-scale datasets and capturing complex spatiotemporal dependencies. These models have demonstrated strong generalization capabilities and offer a scalable, data-driven approach to surrogate modeling [27].

**Semi-supervised learning for Regression.** While SSL has seen significant success in image-related tasks, such as classification and segmentation[51, 67, 58], its use in regression problems remains relatively underexplored [28]. Methods like SSDKL [22] utilize pseudo-labeling along with uncertainty minimization. Additionally, ensemble-based consistency strategies have been proposed to improve regression performance [12]. Other efforts have focused on self-supervised learning methods to learn helpful information from heterogeneous sources [44, 50] or integrating pseudo labels to enhance the accuracy of physics-informed neural networks (PINNs) when solving ordinary differential equations (ODEs) [18, 43]. Noteworthy examples include MPLR [26] and Noisy TS [57], which extend the meta pseudo label framework [47] to handle tabular data and ODE-based regression. However, these approaches often face challenges in generalizing to high-dimensional PDE systems due to the increased complexity of solutions [42]. This indicates an urgent need for more scalable SSL frameworks designed explicitly for PDE-driven problems.

**Uncertainty Estimation.** Uncertainty estimation is critical in deep learning tasks that demand reliable generalization and safety-aware decision [1, 15]. Uncertainty can be broadly categorized into epistemic uncertainty, arising from limited data or model capacity, and aleatoric uncertainty, caused by inherent noise in observations [21]. Bayesian neural networks (BNNs) [6] provide a principled way to model both types [25], but exact inference is often intractable due to computational challenges. Practical approximations like the Monte Carlo (MC) Dropout enable efficient estimation of epistemic uncertainty by sampling stochastic subnetworks at test time [14]. At the same time, ensemble methods aggregate predictions from multiple models to improve robustness and calibration [30, 49, 12]. Recent studies propose latent-space methods that evolve compact representations of system states and their associated uncertainties [63], thereby improving uncertainty quantification through more expressive and structured representation learning.

## B    Supplementary Theoretical Analysis

### B.1    Bayesian Interpretation of MC Dropout

Given a training dataset $\mathcal{D}_l$, the goal of Bayesian inference is to compute the posterior distribution over neural network weights $p(w \mid \mathcal{D}_l)$, which captures epistemic uncertainty about the model parameters. For an evaluation input $x_e \in \mathcal{D}_e$, the predictive distribution of the model output $\hat{y}_e$ is given by marginalizing over this posterior:

$$p(\hat{y}_e \mid x_e, \mathcal{D}_e) = \int p(\hat{y}_e \mid x_e, w) \cdot p(w \mid \mathcal{D}_e) \, dw \tag{14}$$

where $p(\hat{y}_e \mid x_e, w)$ denotes the model output (likelihood) under fixed weights $w$, and $p(w \mid \mathcal{D}_e)$ is the posterior distribution over weights. Since this integral is analytically intractable in deep neural networks, MC Dropout is employed as an efficient approximation method. MC Dropout interprets dropout applied during both training and inference as a variational approximation. It defines a variational distribution $q(w)$ over the weights by randomly zeroing out units with a Bernoulli mask. Formally, for each weight matrix $W_i$, we define:

$$W_i = P_i \cdot \text{diag}(z_i), \quad z_{i,j} \sim \text{Bernoulli}(p_i) \tag{15}$$

where $P_i$ are the learned variational parameters and $z_i$ are dropout masks. This results in a mixture distribution over sub-networks, where each sample corresponds to a different realization of the active units. Under this approximation, the predictive distribution is computed as:

$$p(\hat{y}_e \mid x_e) \approx \int p(\hat{y}_e \mid x_e, w) \cdot q(w) \, dw \approx \frac{1}{T} \sum_{t=1}^{T} p(\hat{y}_e \mid x_e, w_t), \quad w_t \sim q(w) \tag{16}$$

To quantify predictive uncertainty, we estimate the predictive variance by computing the sample variance of the model outputs across $T$ stochastic forward passes, along with the contribution from the assumed model noise. Let $\hat{y}_e^{(t)}$ denote the model output at the $t$-th sample. Then the predictive variance is estimated as:

$$\text{Var}(\hat{y}_e \mid x_e) \approx \tau^{-1}I + \frac{1}{T} \sum_{t=1}^{T} \hat{y}_e^{(t)} \hat{y}_e^{(t)\top} - \left( \frac{1}{T} \sum_{t=1}^{T} \hat{y}_e^{(t)} \right) \left( \frac{1}{T} \sum_{t=1}^{T} \hat{y}_e^{(t)} \right)^{\top} \tag{17}$$

where the first term $\tau^{-1}I$ accounts for the uncertainty under a Gaussian likelihood assumption with model precision $\tau$, and the second term corresponds to the epistemic uncertainty due to model parameters. The overall predictive variance thus combines both sources of uncertainty. The model precision $\tau$ can be derived from the variational framework as a function of weight decay $\lambda_d$, dropout rate $p$, prior length-scale $l$, and dataset size $N$, as follows:

$$\tau = \frac{pl^2}{2N\lambda_d} \tag{18}$$

This provides a principled way to interpret standard dropout training as approximate Bayesian inference, enabling deep models to produce not only predictions but also reliable uncertainty estimates with minimal computational overhead.

## B.2 Proof of Theorem 3.1 and Theorem 3.2

**Theorem3.1.** Assume $\{f_t\}_{t=1}^{\infty}$ satisfies $f_t = f(x; \omega_t)$, where $\omega_t \sim q(\omega)$ is i.i.d. and:

$$\begin{aligned} f_t \in [a, b] \Rightarrow |f_t| \leq M \\ \mu := \mathbb{E}[f_t] \end{aligned} \tag{19}$$

We define the exponential moving average estimate as:

$$\mu_{t+1} = \alpha f_t + (1 - \alpha)\mu_t, \quad \mu_1 = f_1 \tag{20}$$

Then for any time $t$, the EMA estimation satisfies the following mean square error bounds:

$$\mathbb{E}[(\mu_t - \mu)^2] \leq \frac{\alpha}{2 - \alpha} \cdot \text{Var}(f) \tag{21}$$

**Proof:** For $e_t = \mu_t - \mu$, we have:

$$e_{t+1} = (1 - \alpha)e_t + \alpha(f_t - \mu) \tag{22}$$

Square and find the expectation:

$$\mathbb{E}[e_{t+1}^2] = (1 - \alpha)^2 \mathbb{E}[e_t^2] + \alpha^2 \mathbb{E}[(f_t - \mu)^2] \tag{23}$$

and $\mathbb{E}[(f_t - \mu)^2] = \text{Var}(f)$ we have:

$$\mathbb{E}[e_{t+1}^2] \leq (1-\alpha)^2 \mathbb{E}[e_t^2] + \alpha^2 \text{Var}(f) \tag{24}$$

This is a non-homogeneous linear recurrence relation with constant term.

We set $x_t = \mathbb{E}[e_t^2], a = (1-\alpha)^2, b = \alpha^2 \cdot \text{Var}(f)$, then we have:

$$x_{t+1} \leq ax_t + b \tag{25}$$

Expand the first few items:

$$
\begin{aligned}
x_1 &= \text{Var}(f) \\
x_2 &\leq ax_1 + b \\
x_3 &\leq ax_2 + b \leq a(ax_1 + b) + b = a^2 x_1 + ab + b \\
x_4 &\leq a^3 x_1 + a^2 b + ab + b \\
&\vdots \\
x_t &\leq a^{t-1} x_1 + b(1 + a + a^2 + \cdots + a^{t-2}) \\
&= a^{t-1} x_1 + b \cdot \frac{1 - a^{t-1}}{1 - a}
\end{aligned}
\tag{26}
$$

When $t \to \infty, a = (1-\alpha)^2 < 1$ we have:

$$\lim_{t \to \infty} x_t \leq \frac{b}{1-a} = \frac{\alpha^2 \cdot \text{Var}(f)}{1 - (1-\alpha)^2} \tag{27}$$

Therefore we have:

$$\lim_{t \to \infty} \mathbb{E}[e_t^2] \leq \frac{\alpha^2}{2\alpha - \alpha^2} \cdot \text{Var}(f) = \frac{\alpha}{2-\alpha} \cdot \text{Var}(f) \tag{28}$$

The upper bound of its solution can be proved by recurrence inequality as follows:

$$\limsup_{t \to \infty} \mathbb{E}[e_t^2] \leq \frac{\alpha}{2-\alpha} \cdot \text{Var}(f) \tag{29}$$

which we complete the proof.

Beyond the convergence of the predictive mean, we further analyze the consistency of the variance estimator in EMC Dropout. We consider the variance estimate $\hat{\sigma}_t^2$ updated recursively via the EMA formula:

$$\hat{\sigma}_{t+1}^2 = \alpha(f_t - \hat{\mu}_{t+1})^2 + (1-\alpha)\hat{\sigma}_t^2 \tag{30}$$

where $f_t$ denotes the stochastic prediction at time $t$ with dropout, and $\hat{\mu}_{t+1}$ is the EMA mean estimator. Let $\mu = \mathbb{E}[f_t]$ and $\sigma^2 = \mathbb{V}[f_t]$ denote the true predictive mean and variance under the dropout-induced approximate posterior $q(w)$, and define the squared deviation $v_t = (f_t - \mu)^2$. The goal is to quantify the asymptotic error of $\hat{\sigma}_t^2$ with respect to $\sigma^2$. We analyze the mean squared error $\mathbb{E}[(\hat{\sigma}_t^2 - \sigma^2)^2]$ under the assumption that $f_t$ is stationary and ergodic. Following a second-order error decomposition and bounding cross-terms involving the EMA mean error $\epsilon_t = \hat{\mu}_t - \mu$, we derive the following bound:

$$\limsup_{t \to \infty} \mathbb{E}\left[(\hat{\sigma}_t^2 - \sigma^2)^2\right] \leq \frac{\alpha}{2-\alpha} \cdot \text{Var}[(f_t - \mu)^2] \tag{31}$$

This result mirrors the convergence behavior of the EMA mean estimator and confirms that the variance estimate $\hat{\sigma}_t^2$ is asymptotically consistent. The bound depends on the variance of the squared deviation, which reflects the kurtosis of the predictive distribution. As in the case of $\hat{\mu}_t$, the smoothing factor $\alpha \in (0, 1)$ governs the bias-variance trade-off: smaller values yield more stable estimates with slower adaptation, while larger values allow faster response at the cost of higher variance. Equation 31

thus establishes a theoretical guarantee for the reliability of EMC Dropout in approximating the uncertainty captured by standard Monte Carlo dropout, without incurring the cost of repeated forward passes.

**Theorem 3.2.** Generalization Bound under Relative $L_2$ Loss in Semi-Supervised Regression

Let $\mathcal{F}$ be a hypothesis class of neural surrogate models with fixed architecture and bounded parameters. Given a training set consisting of $n_l$ labeled samples and $n_u$ pseudo-labeled samples, then for any $\delta \in (0, 1)$, with probability at least $1 - \delta$, the following inequality holds for any $f(\theta) \in \mathcal{F}$, including $S(\cdot; \theta_S)$:

$$\mathbb{E}_{(x,y_u)} \left[ \mathcal{L}_l(y_u, f(x_u; \theta)) \right] \leq \mathcal{L}_{\text{semi}}(f(\theta)) + 2L \cdot \hat{\mathfrak{R}}_{n_l + n_u}(\mathcal{F}) + c \cdot \sqrt{\frac{\log(1/\delta)}{2(n_l + n_u)}} + \lambda_p \cdot \varepsilon_{\text{pseudo}}^{\text{rel}} \quad (32)$$

where $y_u \in D_u$ is the unlabeled data. $\mathcal{L}_{\text{semi}}(f(\theta))$ is the empirical loss on both labeled and pseudo-labeled data for student model. $\hat{\mathfrak{R}}_{n_l + n_u}(\mathcal{F})$ is the empirical Rademacher complexity of $\mathcal{F}$. $L = \frac{2M}{y_{\min}^2}$ is the Lipschitz constant of the loss function. $c$ is a universal constant. $\lambda_p \in [0, 1]$ is the weight of the pseudo-labeled component. $\varepsilon_{\text{pseudo}}^{\text{rel}} := \mathbb{E}_{x \sim \mathcal{D}_u} \left[ \mathcal{L}_l(y_u, f(x_u; \theta)) \right]$ is the expected relative error of the pseudo-labels.

**Proof:** Let $\mathcal{L}(f(\theta)) := \mathbb{E}_{(x,y_u)} \left[ \mathcal{L}_l(y_u, f(x_u; \theta)) \right]$ denote the expected loss under the data distribution. and let $\hat{\mathcal{L}}(f(\theta)) := \frac{1}{n_l + n_u} \left( \sum_{i=1}^{n_l} \mathcal{L}_l(y_i, f(x_i; \theta)) + \sum_{j=1}^{n_u} \mathcal{L}_u(\hat{y}_u, f(x_j, \theta)) \right)$ denote the empirical loss over the $n_l$ labeled and $n_u$ pseudo-labeled examples. Then we assume the loss function $\mathcal{L}_l =$ is $L$-Lipschitz with respect to $f(x; \theta)$, where $L = \frac{2M}{y_{\min}^2}$ for $|f(x) - y| \leq M$.

From the standard Rademacher generalization bound, we have that with probability at least $1 - \delta$:

$$\mathcal{L}(f(\theta)) \leq \hat{\mathcal{L}}(f(\theta)) + 2L \cdot \hat{\mathfrak{R}}_{n_l + n_u}(\mathcal{F}) + c \cdot \sqrt{\frac{\log(1/\delta)}{2(n_l + n_u)}} \quad (33)$$

Now, the pseudo-labeled loss in $\hat{\mathcal{L}}(f(\theta))$ is computed based on $\tilde{y}_j$ rather than true labels. To account for the potential error introduced by the pseudo-labels, we introduce an additional term, which quantifies the expected relative error between pseudo-labels and the ground truth.

Thus, we obtain Eq.32.

## B.3 Details about gradient for the teacher's update

The traditional optimization process using pseudo tags can be described as follows:

$$\theta_S^{PL} = \underset{\theta_S}{\arg\min} \underbrace{\mathbb{E}_{x_u} [\mathcal{L}(T(x_u; \theta_T), S(x_u; \theta_S))]}_{:= \mathcal{L}_u(\theta_T, \theta_S)} \quad (34)$$

Here, $T$ and $S$ denote the teacher and student models, respectively; $\theta$ represents the model parameters; $\mathcal{L}$ denotes the unlabeled loss function; and $\theta_S^{PL}$ refers to the optimal parameters of the student model obtained via the pseudo-labeling strategy. This process is performed on unlabeled data, as traditional semi-supervised learning frameworks typically combine the supervised loss on labeled data with a consistency loss on unlabeled data.

To update the teacher model, we leverage the student model's performance on labeled data, which can be mathematically quantified as the loss function evaluated on the labeled samples. This relationship can be expressed as follows:

$$\mathbb{E}_{x_l, s_l} \left[ \mathcal{L} \left( y_l, S \left( x_l; \theta_S^{\text{PL}} \right) \right) \right] := \mathcal{L}_l \left( \theta_S^{\text{PL}} \right) \quad (35)$$

This can be interpreted as a functional form where $\theta_T$ serves as the input variable and $\theta_S$ as the optimization parameter. Accordingly, the optimal student parameters can be denoted as $\theta_S^{\text{PL}}(\theta_T)$. The loss in Eq.35 above can be defined as $\mathcal{L}_l(\theta_S^{PL}(\theta_T))$.

In this manner, the meta-modeling process is accomplished: the parameters of one model are regarded as inputs to a functional expression, whereas the parameters of another model act as optimization variables. By minimizing a loss function defined over these variables, the optimal solution can be effectively derived.

Accordingly, updating the teacher model within this framework entails minimizing the loss function specified in Eq.35.

$$\min \mathcal{L}_l(\theta_S^{PL}(\theta_T)),$$
$$\text{where } \theta_S^{PL}(\theta_T) = \underset{\theta_S}{\arg\min} \ \mathcal{L}_u(\theta_T, \theta_S) \tag{36}$$

The argmin operation above is not amenable to gradient-based optimization, as it requires waiting until $\theta_S$ converges to its optimum before proceeding to the next step. This dependency clearly disrupts end-to-end training. To address this issue, this paper proposes a one-step approximation:

$$\theta_S^{PL}(\theta_T) \approx \theta_S - \eta_S \cdot \nabla_{\theta_S} \mathcal{L}_u(\theta_T, \theta_S) \tag{37}$$

Here, the goal becomes:

$$\min_{\theta_T} \ \mathcal{L}_l(\theta_S - \eta_S \cdot \nabla_{\theta_S} \mathcal{L}(\theta_T, \theta_S)) \tag{38}$$

Therefore, if the gradient of $\theta_T$ can be computed for this expression, it becomes feasible to apply gradient descent for end-to-end optimization. Let the above equation be denoted as $\mathcal{L}_F$; the detailed solution process is as follows:

$$\frac{\partial \mathcal{L}_F}{\partial \theta_T} = \frac{\partial}{\partial \theta_T} \mathcal{L}_l \left( y_l, S \left( x_l; \mathbb{E}_{\hat{y}_u^{T_\kappa} \sim T(x_u; \theta_T)} \left[ \theta_S - \eta_S \nabla_{\theta_S} \mathcal{L}_u(\hat{y}_u^{T_\kappa}, S(x_u; \theta_S)) \right] \right) \right) \tag{39}$$

For convenience, we denote the updated parameters of the student model as $\bar{\theta}_S'$:

$$\bar{\theta}_S' = \mathbb{E}_{\hat{y}_u^{T_\kappa} \sim T(x_u; \theta_T)} \left[ \theta_S - \eta_S \nabla_{\theta_S} \mathcal{L}_u \left( \hat{y}_u^{T_\kappa}, S(x_u; \theta_S) \right) \right] \tag{40}$$

Then Eq.39 can thus be rewritten as:

$$\underbrace{\frac{\partial \mathcal{L}_F}{\partial \theta_T}}_{1 \times |T|} = \underbrace{\frac{\partial}{\partial \theta_S} \mathcal{L}_l \left( y_l, S(x_l; \theta_S) \right) \Big|_{\theta_S = \bar{\theta}_S'}}_{1 \times |S|} \cdot \underbrace{\frac{\partial \bar{\theta}_S'}{\partial \theta_T}}_{|S| \times |T|} \tag{41}$$

Here, we denote the shape of $\theta_T$ as $|T| \times 1$ and the shape of $\theta_S$ as $|S| \times 1$, respectively. The left-hand side of the above equation can be efficiently addressed using gradient descent, as the gradient can be approximated by computing the difference between the student model parameters before and after updating on the labeled dataset.

$$\theta_S^* = \theta_S - \eta_S \nabla_{\theta_S} \mathcal{L}_l(y_l, S(x_l; \theta_S)) \tag{42}$$

Therefore, the gradient can be readily approximated by computing the difference between the model parameters $\theta$ before and after the update.

$$\eta_S \nabla_{\theta_S} \mathcal{L}_l(y_l, S(x_l; \theta_S)) = \theta_S - \theta_S^* \tag{43}$$

So now we focus on the right side of the equation:

$$\underbrace{\frac{\partial \bar{\theta}'_S}{\partial \theta_T}}_{|S| \times |T|} = \frac{\partial}{\partial \theta_T} \mathbb{E}_{\hat{y}_u^{T_\kappa} \sim T(x_u; \theta_T)} \left[ \theta_S - \eta_S \nabla_{\theta_S} \mathcal{L}_u(\hat{y}_u^{T_\kappa}, S(x_u; \theta_S)) \right]$$

$$= \frac{\partial}{\partial \theta_T} \mathbb{E}_{\hat{y}_u^{T_\kappa} \sim T(x_u; \theta_T)} \left[ \theta_S - \eta_S \cdot \left( \left. \frac{\partial \mathcal{L}_u(\hat{y}_u^{T_\kappa}, S(x_u; \theta_S))}{\partial \theta_S} \right|_{\theta_S = \theta_S} \right)^\top \right] \qquad (44)$$

we denote $\mathbf{g_{S'}}(\hat{y}_u^{T_\kappa}) := \left. \frac{\partial \mathcal{L}_u(\hat{y}_u^{T_\kappa}, S(x_u; \theta_S))}{\partial \theta_S} \right|_{\theta_S = \theta_S}$ and $\mathbf{g_T} := \frac{\partial \bar{\theta}'_S}{\partial \theta_T}$. $\mathbf{g_{S'}}$ is easily computable via gradient descent, and the equation above becomes:

$$\frac{\partial \bar{\theta}'_S}{\partial \theta_T} = -\eta_S \cdot \frac{\partial}{\partial \theta_T} \mathbb{E}_{\hat{y}_u^{T_\kappa} \sim T(x_u; \theta_T)} \left[ \mathbf{g_{S'}}(\hat{y}_u^{T_\kappa}) \right] \qquad (45)$$

It is important to note that $\mathbf{g_S}$ itself does not depend on $\theta_T$; rather, the dependency arises from the fact that $\hat{y}_u$ is generated through a pseudo-labeling mechanism that is parameterized by the teacher model. This process involves the application of the Leibniz integral rule:

$$\frac{\partial}{\partial \theta_T} \mathbb{E}_{\hat{y}_u^{T_\kappa} \sim T(x_u; \theta_T)} \left[ \mathbf{g_{S'}}(\hat{y}_u^{T_\kappa}) \right] = \frac{\partial}{\partial \theta_T} \sum_{\hat{y}_u} p(\hat{y}_u \mid x_u; \theta_T) \, \mathbf{g_{S'}}(\hat{y}_u^{T_\kappa})$$

$$= \sum_{\hat{y}_u} \frac{\partial}{\partial \theta_T} p(\hat{y}_u \mid x_u; \theta_T) \, \mathbf{g_{S'}}(\hat{y}_u^{T_\kappa})$$

$$= \sum_{\hat{y}_u} p(\hat{y}_u \mid x_u; \theta_T) \frac{\partial}{\partial \theta_T} \log p(\hat{y}_u \mid x_u; \theta_T) \, \mathbf{g_{S'}}(\hat{y}_u^{T_\kappa}) \qquad (46)$$

$$= \mathbb{E}_{\hat{y}_u^{T_\kappa} \sim T(x_u; \theta_T)} \left[ \mathbf{g_{S'}}(\hat{y}_u^{T_\kappa}) \frac{\partial}{\partial \theta_T} \log p(\hat{y}_u^{T_\kappa} \mid x_u; \theta_T) \right]$$

At this point, the partial gradient can be computed using the NLL loss term. For clarity, we now organize the expressions on the left- and right-hand sides of the equation as follows:

$$\frac{\partial \mathcal{L}_F}{\partial \theta_T} = \underbrace{\left. \frac{\partial \mathcal{L}_l(y_l, S(x_l; \bar{\theta}'_S))}{\partial \theta_S} \right|_{\theta_S = \bar{\theta}'_S}}_{1 \times |S|} \cdot \underbrace{\frac{\partial \bar{\theta}'_S}{\partial \theta_T}}_{|S| \times |T|}$$

$$= \eta_S \cdot \underbrace{\left. \frac{\partial \mathcal{L}_l(y_l, S(x_l; \bar{\theta}'_S))}{\partial \theta_S} \right|_{\theta_S = \bar{\theta}'_S}}_{1 \times |S|} \cdot \mathbb{E}_{\hat{y}_u^{T_\kappa} \sim T(x_u; \theta_T)} \left[ \underbrace{\mathbf{g_{S'}}(\hat{y}_u^{T_\kappa})}_{|S| \times 1} \cdot \underbrace{\frac{\partial}{\partial \theta_T} \log p(\hat{y}_u^{T_\kappa} \mid x_u; \theta_T)}_{1 \times |T|} \right]$$

$$(47)$$

The expectation term in the above equation can only be approximated through sampling, which is typically performed over mini-batches. Taking a single sample from the batch as an example, its corresponding gradient is given by:

$$\begin{aligned}
\frac{\partial L_F}{\partial \theta_T} =& \eta_S \cdot \frac{\partial \mathcal{L}_l(y_l, S(x_l; \theta_S'))}{\partial \theta_S} \cdot \left( \left. \frac{\partial \mathcal{L}_u(\hat{y}_u^{T_\kappa}, S(x_u; \theta_S))}{\partial \theta_S} \right|_{\theta_S = \theta_S} \right)^\top \cdot \frac{\partial \log p(\hat{y}_u^{T_\kappa} | x_u; \theta_T)}{\partial \theta_T} \\
=& \underbrace{\eta_S \cdot \left( \nabla_{\theta_S'} \mathcal{L}_l(y_l, S(x_l; \theta_S'))^\top \cdot \nabla_{\theta_S} \mathcal{L}_u(\hat{y}_u^{T_\kappa}, S(x_u; \theta_S)) \right)}_{\text{A scalar} := h} \cdot \nabla_{\theta_T} p(\hat{y}_u^{T_\kappa} | x_u; \theta_T) \qquad (48) \\
\approx& h \cdot \text{NLL}(\hat{y}_u^{T_\kappa}; \theta_T)
\end{aligned}$$

which we complete the proof.

Then we show the Taylor estimation for $h$:

As we know $\theta_S' = \theta_S - \eta$. Let $\eta = \eta_S \nabla_{\theta_S} \mathcal{L}_u(\hat{y}_u^{T_\kappa}, S(x_u; \theta_S))$, according to the Taylor's equation we get:

$$\begin{aligned}
L(\theta_S') = L(\theta_S - \eta) \approx& L(\theta_S) - \eta_S \nabla_{\theta_S} L(\theta_S) \\
=& L(\theta_S) - \eta_S \nabla_{\theta_S} \mathcal{L}_u(\hat{y}_u^{T_\kappa}, S(x_u; \theta_S)) \nabla_{\theta_S} L(\theta_S) \\
=& L(\theta_S) - \eta_S \nabla_{\theta_S} \mathcal{L}_u(\hat{y}_u^{T_\kappa}, S(x_u; \theta_S)) \nabla_{\theta_S} \mathcal{L}_l(s_l, S(x_l)) \\
=& L(\theta_S) - h
\end{aligned} \qquad (49)$$

Therefore, we get the estimation of $h$:

$$h = L(\theta_S) - L(\theta_S') \qquad (50)$$

## B.4 Computational Bottlenecks in Feedback Gradient Computation with a Deterministic Teacher

In the absence of the proposed teacher, the gradient of the feedback loss can be computed directly. However, this leads to substantial computational overhead and proves inefficient under the widely adopted reverse-mode automatic differentiation framework.

Specifically, based on the problem formulation provided in Section 3.2, we derive the explicit expression for directly optimizing the feedback loss:

$$\begin{aligned}
\underbrace{\frac{\partial \mathcal{L}_F}{\partial \theta_T}}_{1 \times |T|} =& \underbrace{\left. \frac{\partial}{\partial \theta_S} \mathcal{L}_l\left(y_l, S(x_l; \theta_S)\right) \right|_{\theta_S = \bar{\theta}_S'}}_{1 \times |S|} \cdot \underbrace{\frac{\partial \bar{\theta}_S'}{\partial \theta_T}}_{|S| \times |T|} \\
=& -\eta_S \cdot \underbrace{\frac{\partial \mathcal{L}_l}{\partial \theta_S'}}_{1 \times |S|} \cdot \underbrace{\frac{\partial}{\partial \theta_T} \left[ \frac{\partial \mathcal{L}_u}{\partial \theta_S} \right]^\top}_{|S| \times |T|}
\end{aligned} \qquad (51)$$

We can further expand the unlabeled loss in Eq.51. For brevity, the subscript $u$ is omitted throughout the following derivation:

$$\begin{aligned}
\mathcal{L}_u(\hat{y}^{T_\kappa}, [\hat{y}^S, \hat{\sigma}^S]) =& \frac{1}{N_{ij}} \sum_i \sum_j \left( \frac{\|\hat{s}_{ij}^{T_\kappa} - \hat{y}_{ij}^S\|_2}{(\hat{\sigma}_{ij}^S)^2 \cdot \|y_{ij}^{T_\kappa} + \kappa\|_2} + \log(\hat{\sigma}_{ij}^S)^2 \right) \\
\frac{\partial \mathcal{L}_u}{\partial \theta_S} =& \frac{1}{N_{ij}} \sum_i \sum_j \left[ \frac{\partial S(x_{ij}; \theta_S)}{\partial \theta_S} \right]
\end{aligned} \qquad (52)$$

Combine Eq.51 and 52, we can get:

$$\underbrace{\frac{\partial \mathcal{L}_F}{\partial \theta_T}}_{1 \times |T|} = -\eta_S \cdot \underbrace{\frac{\partial \mathcal{L}_l}{\partial \theta'_S}}_{1 \times |S|} \cdot \sum_i \sum_j \underbrace{\left[\frac{\partial S(x_{ij}; \theta_S)}{\partial \theta_S}\right]^\top}_{|S| \times 1} \cdot \underbrace{\frac{\partial T(x_{ij}; \theta_T)}{\partial \theta_T}}_{1 \times |T|} \tag{53}$$

The direct computation of Eq.53 requires evaluating gradients at each element in the rollout, leading to prohibitive computational and memory costs. Given a batch of size $B$, with $T$ time steps, $N$ points, state dimensionality $C$, and parameter dimensionality $S$, the memory complexity scales as $\mathcal{O}(B \times T \times N \times C \times S \times T)$, due to the need to preserve the entire computational graph. To overcome these limitations, Section 3.2 introduces UMPL to get interpretable approximation to the feedback gradient. In particular, we adopt the REINFORCE rule [60] to estimate the feedback gradient, which removes the backpropagation path from the student to the teacher and substantially reduces memory and computation, at the expense of longer runtime from repeated forward and backward passes.

## C Training Details

Prior to the formal UMPL training process, we perform a pretraining stage without dropout, yielding an initial model with parameters denoted as $\theta_0$. The teacher is initialized from the warm-up weights $(\theta_T = \theta_{\text{warm}})$, whereas the student is randomly initialized $(\theta_S = \theta_{\text{rand}})$ for the UMPL stage. To enhance training stability and robustness, we construct a memory bank for the unlabeled data $D^u$, storing the mean $\{\mu_0^i\}_{i=1}^{n_u}$ and variance $\{\sigma_0^i\}_{i=1}^{n_u}$ obtained from the teacher's initial stochastic forward passes. This memory bank enables principled estimation of epistemic uncertainty in the teacher network by leveraging the EMC framework, thereby facilitating a more robust distinction between intrinsic model uncertainty and stochastic noise arising during training.

Furthermore, we adopt a Progressive Uncertainty Filtering (PUF) strategy to suppress unreliable pseudo labels. Specifically, at each training step, we compute a binary mask $m \in \{0, 1\}^N$ based on the predictive variance estimated via EMC Dropout. The top-k% most uncertain predictions $(m_i = 1)$ are excluded from both the student's loss computation and the feedback used to update the teacher. As training progresses and pseudo-label quality improves, the masking ratio gradually decreases, allowing the student to incorporate a broader range of samples and provide more informative feedback to guide the teacher's update.

The unified training process consists of the following steps:

1. Get predictions from the teacher:

$$\hat{y}_T^u(\theta_T) = T(x_u; \theta_T) \tag{54}$$

2. Update predicted uncertainty with Eqn. 30, and get uncertainty $\sigma_t^2$.

3. Generate pseudo labels:

$$\hat{y}_u^{T_\kappa} = \hat{y}_u^T(\theta_T) + \xi(g) \cdot \hat{\sigma}_t, \tag{55}$$

4. Calculate the error of the student model on $D^l$ and Train the student with unlabeled loss ($\mathcal{L}_S^u$):

$$\begin{aligned}
\mathcal{L}_l^S &= \mathcal{L}_l(y_l, S(x_l, \theta_S)) \\
\mathcal{L}_u^S &= \mathcal{L}_u(m \cdot \hat{y}_u^{T_\kappa}, m \cdot S(x_u, \theta_S)) \\
\theta'_S &= \theta_S - \eta_S \cdot \nabla_{\theta_S} \mathcal{L}_u^S
\end{aligned} \tag{56}$$

5. Calculate the error of the updated student model on $D^l$ and feedback loss with :

$$\begin{aligned}
\mathcal{L}_l^{S'} &= \mathcal{L}_2(y_l, S'(x_l, \theta_{S'})) \\
\mathcal{L}_F^l &:= \mathcal{L}_S^l - \mathcal{L}_{S'}^l
\end{aligned} \tag{57}$$

6. Train the teacher with both labeled loss $\mathcal{L}_T^l$ and the feed back loss $\mathcal{L}_F^l$:

$$
\begin{aligned}
\mathcal{L}_T^l &= \mathcal{L}_l(y_l, T(x_l, \theta_T)) \\
\theta_T' &= \theta_T - \eta_T \cdot \nabla_{\theta_T} \cdot (\mathcal{L}_T^l + \mathcal{L}_F^l)
\end{aligned}
\tag{58}
$$

To ensure stability throughout training, we apply a **Supervised Anchoring (SA)** strategy where the student is consistently trained on labeled data after the last step, regardless of pseudo-label reliability. This mitigates error propagation during self-training and is especially crucial under distribution shifts where pseudo-labels may be misleading. Beyond SA, our algorithm minimizes model-collapse risk by coupling divergent teacher-student initialization and independently sampled labeled anchors with an uncertainty-guided feedback loop (student's aleatoric signals) and uncertainty-aware pseudo-labeling (teacher's epistemic perturbation with progressive uncertainty filtering), curbing error reinforcement and stabilizing convergence.

# D    Details of Compared Approaches

**Pseudo Labeling(PL)[31].** In this baseline, the teacher model is fixed throughout training and is used to generate pseudo labels for the unlabeled data. The student model is then trained jointly on the labeled data and the pseudo-labeled unlabeled data with a supervised loss. While this method leverages additional information from the unlabeled set, our experiments reveal that it can underperform compared to fully supervised training. This performance degradation is primarily attributed to the teacher's inaccurate or noisy predictions on the unlabeled samples, which may introduce erroneous learning signals and ultimately misguide the student model.

**Mean Teacher[54].** The Mean Teacher framework improves upon standard pseudo-labeling by dynamically updating the teacher model as an exponential moving average (EMA) of the student's parameters during training. At each iteration, the teacher generates pseudo labels for the unlabeled data. At the same time, the student is trained to minimize both the supervised loss of labeled data and the consistency loss concerning the teacher's predictions. Despite its stability advantages, Mean Teacher has a similar drawback to standard pseudo-labeling. If the teacher's predictions on unlabeled data are inaccurate, they can propagate noise and mislead the student. In our experiments, we set the EMA decay coefficient to 0.99, which balances temporal smoothness and adaptation.

**Noisy Teacher-Student (TS)[57].** This method extends the standard teacher-student framework by introducing stochastic perturbations to the teacher model and incorporating feedback from the student to guide the generation of pseudo labels. Specifically, the teacher produces rollouts under added Gaussian noise, which are then used to train the student model. The student's performance on labeled data provides a feedback signal to update the teacher, helping to mitigate the accumulation of errors caused by inaccurate or noisy pseudo labels. This feedback mechanism improves the student's stability and prediction accuracy, particularly in low-label regimes. In our implementation, the noise magnitude applied to the teacher's outputs is fixed at 0.05, following the original formulation.

# E    Detailed description of datasets, models, and experimental settings

## E.1    Datasets

Table 5: Basic information of datasets. Note that No. train includes both labeled and unlabeled data.

| Dataset | Domain | # Nodes(avg.) | Timesteps | Operator Mapping | # Phys. | # No. train/test |
|---------|--------|---------------|-----------|------------------|---------|------------------|
| Darcy Flow | 2D | 2290 | / | $a(\mathbf{x}) \mapsto u(x)$ | $u = [U]$ | 1000/200 |
| Stationary Lid | 2D | 1475 | / | BC.+Param $\mapsto u(x)$ | $u = [U, V, P]$ | 861/100 |
| Time-dependent Lid2d | 2D+time | 930 | 101 | IC.+Param $\mapsto u(x,T)$ | $u = [U, V, P]$ | 900/100 |
| NSM2d | 2D+time | 2630 | 201 | IC.+Param $\mapsto u(x,T)$ | $u = [U, V, P]$ | 900/100 |
| Black Sea | 2D+time | 5000 | 15 | IC.+Param $\mapsto u(x,T)$ | $u = [T]$ | 1615/238 |
| Plasma ICP | 2D+time | 3424 | 81 | IC.+Param $\mapsto u(x,T)$ | $u = [n_e, \phi, T]$ | 500/150 |
| Ahmed | 3D | 110k | / | Geo.+Param $\mapsto C_d$ | / | 500/51 |

As shown in Table 5, we employ seven datasets spanning diverse domains, such as fluid dynamics, plasma ,and large-scale 3D geometry simulations. A summary of the datasets, their underlying operator learning tasks, and data generation sources is provided below.

**Darcy Flow.** We adopt the Irregular Darcy dataset [9], which is based on the Darcy flow equation over a 2D irregular domain with a thin rectangular notch. The governing equation is $-\nabla \cdot (a\nabla u) = f$, where the source term is fixed to $f = 1$. The input diffusion field $a(\mathbf{x})$ is generated by a Gaussian random field $\mu \sim \mathcal{N}(0, (-\Delta + 25I)^{-2})$, followed by a piecewise transformation $t(\mu)$ to produce binary-valued coefficients: $t(\mu) = 12$ if $\mu \geq 0$, and 4 otherwise. The computational mesh contains 2290 nodes with Dirichlet boundary conditions. The dataset provides 1200 labeled samples, of which 1000 are used for training and 200 for testing.

**Stationary Lid & Time-dependent Lid2d.** The Lid2d dataset is based on the lid-driven cavity problem, a canonical benchmark in computational fluid dynamics that characterizes recirculating flow induced by a moving top boundary (the "lid") within a square domain. A constant horizontal velocity is applied at the lid, while the remaining three walls follow no-slip boundary conditions. We construct two versions of this problem: a stationary (steady-state) version and a time-dependent (transient) version, each governed by different formulations of the incompressible Navier-Stokes equations.

*Stationary Lid2d:* In the stationary case, the flow is assumed to have fully developed and no longer changes over time. Therefore, the governing equation reduces to the steady-state Navier-Stokes form:

$$
\begin{cases}
-\boldsymbol{u} \cdot \nabla \boldsymbol{u} + \dfrac{1}{Re}\Delta \boldsymbol{u} - \nabla p = 0, \\
\qquad\qquad\qquad\quad \nabla \cdot \boldsymbol{u} = 0,
\end{cases}
\tag{59}
$$

where $\boldsymbol{u} = (u(x,y), v(x,y))$ is the velocity field, $p(x,y)$ is the pressure, and Re is the Reynolds number. The spatial domain is discretized using a non-uniform triangular mesh with 1475 nodes. The steady-state dataset is generated over a uniform grid of $Re \in [100, 10000]$ and lid velocity $v \in [0.1, 1.0]$, forming a 2D parameter grid of $31 \times 31$ samples. Among them, 81 samples centered in the grid are selected for training, 100 are randomly chosen for testing, and the remaining 780 are used as unlabeled data.

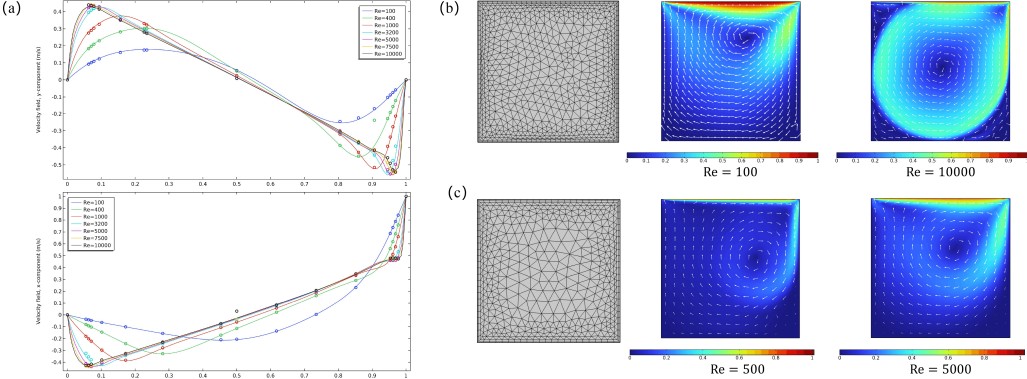

Figure 5: Visualization of the Lid2d dataset. (a) Velocity profiles along $x = 0.5$ and $y = 0.5$ in the stationary Lid dataset. The hollow circles represent reference values from the literature [16], showing excellent agreement with our simulation results. (b)-(c) Mesh structure and velocity magnitude $v = \sqrt{v_x^2 + v_y^2}$ visualization for the stationary and time-dependent Lid2d datasets, respectively. For the time-dependent case, the final time step is visualized.

*Time-dependent Lid2d.* In contrast, the time-dependent case follows the full unsteady Navier-Stokes system:

$$
\begin{cases}
\dfrac{\partial \boldsymbol{u}}{\partial t} = -\boldsymbol{u} \cdot \nabla \boldsymbol{u} + \dfrac{1}{Re}\Delta \boldsymbol{u} - \nabla p, \\
\nabla \cdot \boldsymbol{u} = 0.
\end{cases}
\tag{60}
$$

subject to the initial condition $(u, v, p) = \mathbf{0}$ and the same boundary conditions as in the stationary case. Simulations are conducted over a non-uniform triangular mesh with 730 nodes, spanning a total simulation time of 10 seconds with 101 time steps. We consider two Reynolds numbers $(Re = 500, Re = 5000)$ and uniformly sample lid velocities $v \in [0.1, 1.0]$. For each Re, 1000 transient trajectories are generated. Among these, 100 samples are fixed for testing, while the remaining 900 are treated as unlabeled data for semi-supervised learning. Visualization about this dataset is shown in Figure5.

**NSM2d.** The NSM2d dataset is derived from the 2D incompressible Navier-Stokes equations with an additional magnetic force term. The simulation follows the same physical setting as the original reference [37], including the time-invariant magnetic intensity, inflow velocity profile, and boundary conditions. The domain is $[0, 4] \times [0, 1]$, with a jet inflow imposed at $x = 0$, centered at a vertical position $y_0 \in [0.4, 0.6]$, and Reynolds numbers ranging from 100 to 1500. While preserving the original PDE formulation, we discretize the spatial domain using a non-uniform triangular mesh with 2630 nodes. One thousand simulations are generated using COMSOL, each consisting of 201 time steps over a 10-second interval. A visualization of the dataset is shown in Figure6.

(a)                                    (b)

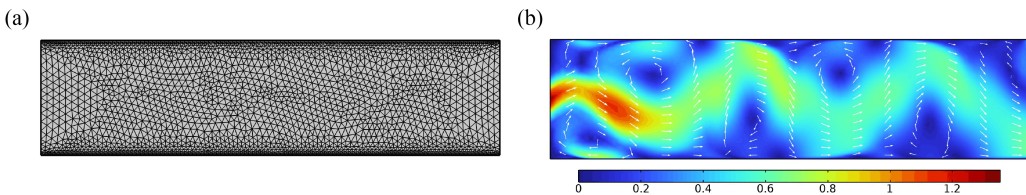

Figure 6: Visualization of the NSM2d dataset. (a) Mesh of the NSM2d dataset. (b) Visualization of the velocity magnitude $v = \sqrt{v_x^2 + v_y^2}$ at the final time step.

**Black Sea.** The Black Sea dataset comprises daily real-world measurements of ocean currents and temperatures [35]. We use data from $01/01/2017$ to $01/01/2022$. The raw data is provided on a regular grid with a spatial resolution of $1/27° \times 1/36°$, but due to the irregular geometry of the sea, fewer than 50% of grid points lie within the valid ocean region. We apply Delaunay triangulation to the valid region to construct a mesh, resulting in a spatial discretization with 5001 nodes consistent across all models. Following [20], we use the water temperature at a fixed depth of 12.54 meters as the predictive variable.

For model evaluation, we adopt a temporal extrapolation and interpolation split. Specifically, we designate the final three months of the dataset as the extrapolation test set. In comparison, three additional non-consecutive months are randomly selected from the earlier portion of the dataset as the interpolation test set. The remaining data is used for training under a semi-supervised setting. During training, the model takes the temperature field of a single day as input and is trained to predict the temperature evolution over the next 5 consecutive days. For testing, the model is evaluated on its ability to forecast the next 14 days, given only the temperature from a single input day. The dataset is available to download from the Copernicus Institute[3]. The geometry of this problem and the solution field are shown in the following Figure 7.

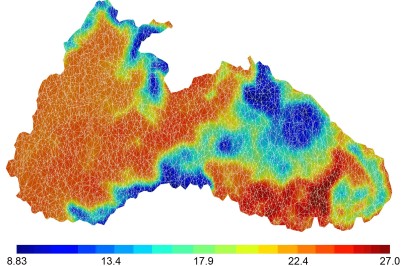

Figure 7: Visualization of the Black Sea dataset with mesh and the solution field $T$.

---

[3]Available at https://data.marine.copernicus.eu/product/BLKSEA_MULTIYEAR_PHY_007_004/description

**Plasma ICP.** This dataset models an argon/oxygen inductively coupled plasma reactor, incorporating multiphysics phenomena such as magnetic fields, plasma dynamics, laminar flow, and heat transfer. The model is derived from the COMSOL Application Library[4], with partial chemical reaction mechanisms adopted from [48]. To construct a multi-fidelity learning scenario, we generate two levels of simulation fidelity:

- Low-fidelity simulations are conducted on a sparser mesh with approximately 10,000 elements and exclude specific physical and chemical processes.
- High-fidelity simulations use a denser mesh with around 25,000 elements and preserve the complete physical mechanisms, providing more accurate but computationally expensive results (about 1 hour per sample for computing).

Varying reactor conditions, including chamber pressure, inlet flow rate, argon/oxygen ratio, and chuck temperature, generate 650 high-fidelity simulations. Among them, 500 samples are used as labeled or unlabeled data during training, while the remaining 150 samples are reserved exclusively for testing under distribution shifts. The training protocol follows a standard multi-fidelity learning pipeline: models are first pretrained on the low-fidelity dataset, which provides a broader but less detailed approximation of the physical system. This is followed by fine-tuning and evaluating the high-fidelity dataset, enabling the assessment of model generalization across fidelity gaps under limited supervision. Each simulation spans a total physical time of 1 second, discretized using 81 non-uniform time steps. The time stepping follows a logarithmic scale, specifically $t = 10^{\text{linspace}(-8,\, 0.1,\, 0)}$, better to capture both early fast transients and slower dynamics. A visual comparison between low- and high-fidelity solutions is presented in Figure 8, highlighting differences in mesh resolution and solution fields.

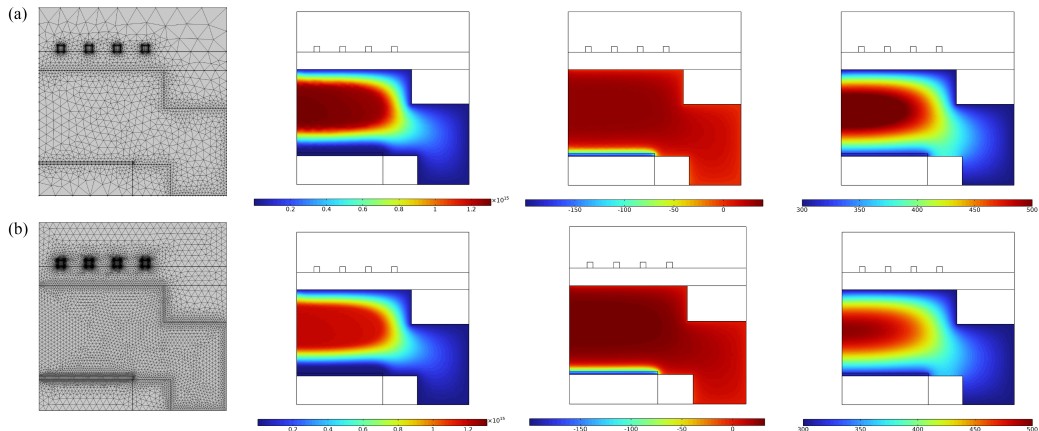

Figure 8: A visual comparison between low- and high-fidelity mesh and solutions. (a) shows the low-fidelity data, and (b) the high-fidelity data. From left to right, each panel presents the mesh, electron density ($n_e$), electric potential ($\phi$), and gas temperature ($T$), highlighting the differences in both mesh resolution and solution fields across fidelities.

**Ahmed.** The Ahmed Body dataset is constructed from parametric variations of a simplified vehicle geometry, characterized by six shape parameters—length, width, height, ground clearance, slant angle, and fillet radius—along with the inlet velocity to account for changes in Reynolds number. 551 design configurations are generated using Latin Hypercube Sampling, with 500 for training and 51 for testing. Aerodynamic simulations are performed using OpenFOAM with the SST $k$-$\omega$ turbulence model. The resulting surface pressure distributions are used to compute the drag coefficient $C_d$ based on the following standard integral:

$$C_d = \frac{2}{v^2 A} \int_S p(x)\,(\mathbf{n}(x) \cdot \mathbf{i})\,dx, \tag{61}$$

where $v$ is the inlet velocity, $A$ is the frontal area, $p(x)$ is the surface pressure, $\mathbf{n}(x)$ is the surface normal, and $\mathbf{i}$ is the streamwise direction. Details of the geometry and pressure computation follow

---

[4]Available at https://www.comsol.com/model/model-of-an-argon-oxygen-inductively-coupled-plasma-reactor-109191

standard aerodynamic practices and can be found at Modulus Datasets[5]. This dataset defines a mapping from vehicle geometry and flow conditions to aerodynamic drag, using the surface mesh (about 110k nodes) as model input. A visualization of the mesh and pressure is shown in Figure 9.

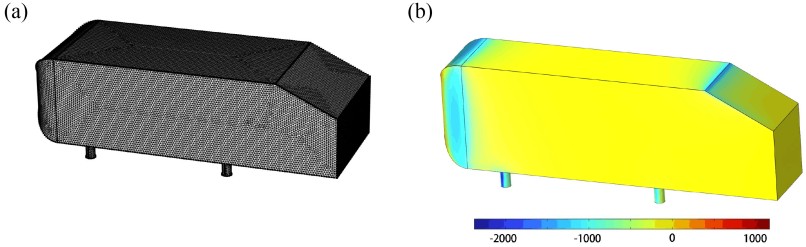

Figure 9: Visualization of the mesh (a) and surface pressure (b) for the Ahmed dataset.

## E.2 Models

This section presents the models employed in our experiments, including two point-cloud-based architectures (DeepONet[41] and Transolver[61]) and two mesh-based architectures (MeshGraphNet (MGN)[46]. Aerogto[38]). To ensure a fair comparison, we retain the original network structures for all approaches without modification. All models are trained and evaluated autoregressively for temporal prediction tasks, where future states are recursively predicted based on previous outputs. In addition to the primary output, we estimate data-dependent uncertainty by attaching an auxiliary decoder to each model, following the model-based method[63]. This auxiliary decoder learns to predict the predictive variance conditioned on the input data, allowing the models to provide uncertainty estimates alongside pointwise predictions.

**DeepONet[41].** DeepONet is a neural operator architecture composed of a branch network that encodes the input function and a trunk network that encodes spatial coordinates. In our experiments, both networks consist of three fully connected layers with GELU activations and 128 hidden units. The outputs of the two networks are combined via element-wise addition and passed through a two-layer output network to produce the final prediction. We train the model using the Adam optimizer, with a cosine annealing learning rate schedule that decays from $1e^{-3}$ to $1e^{-5}$ over the course of training epochs.

**Transolver[61].** Transolver is a transformer-based architecture tailored for learning physical dynamics on irregular spatial meshes. It represents mesh nodes as unordered sequences and employs a specialized slice-token attention mechanism to capture both local and global interactions. In our implementation, Transolver consists of 8 transformer layers with 8 attention heads, a hidden dimension of 128, and a slice-token mechanism with 64 slices. The input combines node coordinates and physical state variables, optionally augmented with time embeddings for temporal predictions. All features are first processed through an MLP and added to a learnable positional placeholder before passing through the transformer blocks. The model is trained with the AdamW optimizer, with an initial learning rate of $5e^{-4}$ and a weight decay of $1e^{-5}$.

**MeshGraphNet(MGN)[46].** MGN is a graph neural network designed to learn physical dynamics over unstructured meshes. It models each simulation domain as a graph, where nodes represent mesh points and edges encode spatial relationships. The architecture consists of an encoder that embeds node and edge features, a processor composed of 12 message-passing steps to propagate information across the mesh, and a decoder that maps the final node embeddings to predicted physical states. All MLPs in the model use 2 hidden layers, 128-dimensional embeddings, ReLU activations, layer normalization, and dropout for regularization. In our experiments, MGN is trained with the Adam optimizer with a learning rate that decays from $1e^{-3}$ to $1e^{-5}$.

**AeroGTO[38].** AeroGTO is a hybrid graph-transformer architecture designed for learning aerodynamic responses over unstructured 3D meshes. It combines local message passing via a graph neural network (GNN) with global correlation modeling through a projection-based attention mechanism

---

[5]Available at https://catalog.ngc.nvidia.com/orgs/nvidia/teams/modulus/resources/modulus_datasets-ahmed_body_test

(GTO-Atten). The model encodes node and edge features using MLPs with Fourier-based positional encodings. It processes them through a sequence of 4 Mixer blocks, each composed of a GNN and a transformer-style attention layer with 128 learnable tokens and 4 attention heads. With a hidden dimension 128, AeroGTO captures fine-grained geometric structures and long-range dependencies across the mesh. The final predictions are produced through a residual decoder with a learnable scaling factor. The model is trained using the AdamW optimizer, with a learning rate that decays from $1e^{-4}$ to $1e^{-6}$.

### E.3 Experimental Details

This section outlines the experimental settings, including training schedules, data splits, and task-specific configurations.

**Training Schedule.** We establish a fully supervised baseline ("Supervised Only") for each task type, with training epochs determined by task complexity. Specifically, stationary tasks are trained for 500 epochs, transient tasks for 1000 epochs, and the Ahmed dataset is trained for 300 epochs. In the case of distribution shift experiments, models are first pretrained on source-domain data for 1000 epochs, followed by fine-tuning on a small, shifted-domain subset for an additional 300 epochs. Semi-supervised methods, including Pseudo Labeling (PL), Mean Teacher, Moisy Teacher-Student, and UMPL, follow the same total training schedule as their corresponding supervised baselines. Semi-supervised training begins from the supervised initialization and continues for the same number of epochs.

**Labeled-Unlabeled Split.** For semi-supervised learning, we randomly select a fixed proportion $r \in [10\%, 30\%]$ of the training data to serve as labeled samples, while the remaining data is treated as unlabeled. The Stationary Lid dataset is an exception, where 81 samples are consistently selected as the labeled subset, and all other samples are used as unlabeled data.

**Distribution Shifts Setup.** We adopt a two-stage training strategy to evaluate model robustness under distribution shifts involving pretraining and fine-tuning across distinct data distributions. Specifically, in the Time-dependent Lid2d dataset, we perform cross-Reynolds transfer, where models pretrained on $Re = 500$ are fine-tuned on data from $Re = 5000$, and vice versa, simulating a distribution shift in flow regimes. In the Plasma ICP dataset, we evaluate fidelity transfer, where models are pretrained on low-fidelity simulations and then fine-tuned on high-fidelity data to assess generalization across simulation accuracies. In both cases, pretraining is conducted for 1000 epochs on the source domain, followed by fine-tuning for 300 epochs on a small, labeled subset ($r \in [10\%, 30\%]$) of the target domain.

**UMPL Settings.** For all datasets, UMPL training is conducted with a fixed dropout rate of 0.1, $\alpha = 0.2$, and an initial top-$k$ rate of 0.5. The dropout rate refers to the probability used in MC estimation of the teacher model, controlling the stochasticity during inference. The parameter $\alpha$ governs the EMA update of the teacher model, balancing current student feedback with past teacher parameters. The initial top-$k$ rate determines the mask ratio used for pseudo-label selection: at the beginning of training, only the top-$k$ percent of predictions with the highest confidence are selected for student learning and feedback, while the rest are masked out to mitigate the impact of noisy supervision.

## F More Experiment Results

This section presents additional experimental results, including extended comparison studies and detailed ablation analyses.

### F.1 Computational Cost Analysis

All experiments were conducted on a single NVIDIA RTX 4090 GPU. We evaluate the computational efficiency of the proposed method using the AeroGTO model across three representative tasks-2D steady-state, 2D transient, and 3D simulation. The computation time and memory consumption for each case are summarized in 6 to provide a fair comparison of the overall computational cost.

Here, Supervised+uq refers to the student model used in UMPL. As shown, UMPL introduces almost no additional memory overhead, thanks to the REINFORCE-style update [60] that avoids the costly

Table 6: Computation time and memory usage across datasets.

| Method | Darcy (bs=4) | | NSM2d (bs=4) | | Ahmed (bs=1) | |
|---|---|---|---|---|---|---|
| | Memory (M) | Training Time (h) | Memory (M) | Training Time (h) | Memory (M) | Training Time (h) |
| Supervised | 2297 | 0.45 | 9727 | 3.75 | 9375 | 3.53 |
| Pseudo Label | 3843 | 1.08 | 16567 | 9.04 | 15947 | 8.73 |
| Mean Teacher | 3843 | 1.21 | 16567 | 10.17 | 15947 | 9.57 |
| Noisy TS | 2304 | 1.77 | 9813 | 13.21 | 9477 | 12.48 |
| Supervised+UQ | 3243 | 0.96 | 13578 | 7.69 | 12986 | 7.36 |
| UMPL | 3257 | 2.21 | 13792 | 16.48 | 13290 | 15.56 |

Table 7: Comparison of the NSM2d and Time-dependent Lid2d dataset across different methods and models. **Cyan**, **Yellow**, and **Green** indicate the best, second-best, and worst $L_2$ loss values, respectively.

| Model | Method | BENCHMARKS | | | | | | | | |
|---|---|---|---|---|---|---|---|---|---|---|
| | | NSM2d | | | Time-dependent Lid2d ($Re = 500$) | | | Time-dependent Lid2d($Re = 5000$) | | |
| | | 10% | 20% | 30% | 10% | 20% | 30% | 10% | 20% | 30% |
| DeepONet | Supervision Only | 3.07e-1 | 2.54e-1 | 2.24e-1 | 1.81e-1 | 1.43e-1 | 1.27e-1 | 4.88e-1 | 3.34e-1 | 3.06e-1 |
| | Pseudo Label | 2.82e-1 | 2.43e-1 | 2.09e-1 | 2.19e-1 | 1.40e-1 | 1.27e-1 | 5.02e-1 | 3.15e-1 | 2.63e-1 |
| | Mean Teacher | 2.77e-1 | 2.39e-1 | 2.02e-1 | 2.03e-1 | 1.33e-1 | 1.12e-1 | 4.91e-1 | 3.08e-1 | 2.55e-1 |
| | Noisy TS | 2.57e-1 | 2.24e-1 | 1.73e-1 | 1.46e-1 | 1.22e-1 | 1.02e-1 | 2.79e-1 | 2.43e-1 | 2.04e-1 |
| | UMPL(ours) | 2.23e-1 | 1.91e-1 | 1.52e-1 | 1.29e-1 | 1.06e-1 | 8.93e-2 | 2.51e-1 | 2.12e-1 | 1.87e-1 |
| | **PROMOTION** | **13.22%** | **14.73%** | **12.13%** | **11.64%** | **13.11%** | **10.49%** | **10.03%** | **12.75%** | **8.33%** |
| MGN | Supervision Only | 5.62e-1 | 3.46e-1 | 1.76e-1 | 2.79e-1 | 1.37e-1 | 9.78e-2 | 3.21e-1 | 1.90e-1 | 8.17e-2 |
| | Pseudo Label | 4.95e-1 | 2.94e-1 | 1.54e-1 | 2.72e-1 | 1.26e-1 | 7.76e-2 | 3.14e-1 | 1.71e-1 | 6.47e-2 |
| | Mean Teacher | 4.41e-1 | 2.79e-1 | 1.33e-1 | 2.71e-1 | 1.15e-1 | 5.58e-2 | 3.10e-1 | 1.61e-1 | 6.09e-2 |
| | Noisy TS | 3.85e-1 | 1.99e-1 | 1.07e-1 | 2.55e-1 | 9.60e-2 | 4.45e-2 | 3.04e-1 | 1.33e-1 | 4.74e-2 |
| | UMPL(ours) | 3.51e-1 | 1.61e-1 | 8.03e-2 | 2.36e-1 | 8.02e-2 | 2.82e-2 | 2.93e-1 | 1.12e-1 | 4.48e-2 |
| | **PROMOTION** | **8.83%** | **19.09%** | **24.95%** | **7.59%** | **16.30%** | **36.60%** | **11.37%** | **16.38%** | **5.48%** |
| Transolver | Supervision Only | 2.71e-1 | 1.29e-1 | 5.41e-2 | 2.15e-1 | 7.15e-2 | 3.65e-2 | 2.76e-1 | 1.17e-1 | 8.89e-2 |
| | Pseudo Label | 2.35e-1 | 1.17e-1 | 5.04e-2 | 2.12e-1 | 6.86e-2 | 3.23e-2 | 2.70e-1 | 1.12e-1 | 8.53e-2 |
| | Mean Teacher | 2.21e-1 | 1.15e-1 | 4.99e-2 | 2.10e-1 | 6.58e-2 | 3.11e-2 | 2.57e-1 | 1.10e-1 | 8.18e-2 |
| | Noisy TS | 1.79e-1 | 8.85e-2 | 4.00e-2 | 2.09e-1 | 6.38e-2 | 2.96e-2 | 2.54e-1 | 1.09e-1 | 8.04e-2 |
| | UMPL(ours) | 1.51e-1 | 7.50e-2 | 3.83e-2 | 2.04e-1 | 5.89e-2 | 2.53e-2 | 2.40e-1 | 1.05e-1 | 7.80e-2 |
| | **PROMOTION** | **15.64%** | **15.25%** | **4.25%** | **2.39%** | **7.68%** | **14.52%** | **5.51%** | **3.66%** | **2.98%** |
| AeroGTO | Supervision Only | 1.72e-1 | 9.35e-2 | 5.13e-2 | 5.02e-2 | 2.23e-2 | 1.21e-2 | 9.46e-2 | 4.84e-2 | 2.66e-2 |
| | Pseudo Label | 1.51e-1 | 9.04e-2 | 4.54e-2 | 5.37e-2 | 1.91e-2 | 1.03e-2 | 7.66e-2 | 4.20e-2 | 2.26e-2 |
| | Mean Teacher | 1.45e-1 | 8.71e-2 | 4.29e-2 | 4.59e-2 | 1.41e-2 | 9.30e-3 | 7.05e-2 | 3.55e-2 | 1.90e-2 |
| | Noisy TS | 1.39e-1 | 8.22e-2 | 3.96e-2 | 4.24e-2 | 1.20e-2 | 6.89e-3 | 6.34e-2 | 3.06e-2 | 1.75e-2 |
| | UMPL | 1.27e-1 | 7.54e-2 | 3.69e-2 | 3.51e-2 | 1.22e-2 | 5.18e-3 | 5.74e-2 | 2.57e-2 | 1.61e-2 |
| | **PROMOTION** | **8.63%** | **8.27%** | **6.81%** | **17.20%** | **7.04%** | **24.71%** | **9.39%** | **15.69%** | **7.83%** |

backpropagation chain from the student to the teacher. This significantly reduces computational and memory complexity, at the cost of increased training time.

## F.2 Extended Comparison Results

We conduct further comparisons across different models and various proportions of labeled data. Table 7 reports the performance of different methods and models on the NSM2d and Time-dependent Lid2d datasets under varying amounts of labeled data. Table 8 summarizes results on the Black Sea dataset, comparing methods both in temporal interpolation (In-T) and extrapolation (Out-T) scenarios. Table 9 presents a similar comparison of the Plasma ICP dataset. Across all these tasks, UMPL consistently outperforms baseline methods, demonstrating its effectiveness under limited supervision.

Figure 10(a) shows raincloud plots of prediction versus ground truth on the Ahmed dataset, revealing that UMPL yields significantly better generalization on out-of-distribution(OOD) test samples than baseline approaches. Figure 10(b) illustrates the training curves on the Stationary Lid dataset. While

Table 8: Comparison of the Black Sea dataset across different methods and models. **Cyan** , **Yellow** , and **Green** indicate the best, second-best, and worst $L_2$ loss values, respectively.

| Model | Method | Black Sea | | | | | |
|---|---|---|---|---|---|---|---|
| | | 10% | | 20% | | 30% | |
| | | In-T | Out-T | In-T | Out-T | In-T | Out-T |
| DeepONet | Supervision Only | 1.45e-1 | 2.41e-1 | 1.10e-1 | 1.23e-1 | 1.06e-1 | 9.26e-2 |
| | Pseudo Label | 1.51e-1 | 2.35e-1 | 1.02e-1 | 1.14e-1 | 1.02e-1 | 9.07e-2 |
| | Mean Teacher | 1.45e-1 | 2.22e-1 | 9.61e-2 | 1.07e-1 | 9.98e-2 | 8.89e-2 |
| | Noisy TS | 1.24e-1 | 1.67e-1 | 8.61e-2 | 9.63e-2 | 9.81e-2 | 8.06e-2 |
| | UMPL(ours) | 9.61e-2 | 1.43e-1 | 7.97e-2 | 8.70e-2 | 9.09e-2 | 7.09e-2 |
| | **PROMOTION** | **22.50%** | **14.37%** | **7.43%** | **9.65%** | **7.33%** | **12.03%** |
| MGN | Supervision Only | 9.84e-2 | 1.13e-1 | 8.72e-2 | 9.83e-2 | 7.31e-2 | 6.92e-2 |
| | Pseudo Label | 9.97e-2 | 1.08e-1 | 8.19e-2 | 9.11e-2 | 6.57e-2 | 6.876e-2 |
| | Mean Teacher | 9.83e-2 | 1.08e-1 | 7.91e-2 | 9.05e-2 | 6.41e-2 | 6.73e-2 |
| | Noisy TS | 8.51e-2 | 9.67e-2 | 7.13e-2 | 8.29e-2 | 5.77e-2 | 6.24e-2 |
| | UMPL(ours) | 8.39e-2 | 9.59e-2 | 6.52e-2 | 7.61e-2 | 5.29e-2 | 5.83e-2 |
| | **PROMOTION** | **1.41%** | **0.83%** | **8.55%** | **8.20%** | **8.31%** | **6.52%** |
| Transolver | Supervision Only | 9.34e-2 | 1.11e-1 | 7.56e-2 | 7.99e-2 | 5.49e-2 | 7.07e-2 |
| | Pseudo Label | 9.16e-2 | 9.79e-2 | 7.29e-2 | 7.65e-2 | 4.70e-2 | 6.75e-2 |
| | Mean Teacher | 9.26e-2 | 9.68e-2 | 7.19e-2 | 7.48e-2 | 5.69e-2 | 6.76e-2 |
| | Noisy TS | 8.81e-2 | 9.17e-2 | 6.78e-2 | 6.94e-2 | 5.01e-2 | 6.12e-2 |
| | UMPL(ours) | 8.74e-2 | 9.13e-2 | 6.26e-2 | 6.82e-2 | 4.47e-2 | 5.65e-2 |
| | **PROMOTION** | **1.35%** | **0.43%** | **7.66%** | **1.73%** | **10.77%** | **7.68%** |
| AeroGTO | Supervision Only | 8.95e-2 | 1.10e-1 | 7.08e-2 | 7.46e-2 | 5.11e-2 | 6.85e-2 |
| | Pseudo Label | 8.24e-2 | 1.03e-1 | 6.98e-2 | 7.43e-2 | 4.81e-2 | 6.19e-2 |
| | Mean Teacher | 9.07e-1 | 1.02e-1 | 6.79e-2 | 7.17e-2 | 4.92e-2 | 6.15e-2 |
| | Noisy TS | 7.42e-2 | 9.38e-2 | 6.09e-2 | 7.04e-2 | 4.53e-2 | 5.67e-2 |
| | UMPL | 6.66e-2 | 8.72e-2 | 5.48e-2 | 6.28e-2 | 4.13e-2 | 5.02e-2 |
| | **PROMOTION** | **10.29%** | **7.05%** | **9.92%** | **10.78%** | **8.76%** | **11.55%** |

Table 9: Comparison of the Plasma ICP dataset across different methods and models. **Cyan** , **Yellow** , and **Green** indicate the best, second-best, and worst $L_2$ loss values, respectively.

| Model | Percentage | Method | | | | | | |
|---|---|---|---|---|---|---|---|---|
| | | w/o Fine-tuning | Supervision Only | Pseudo Label | Mean Teacher | Noisy TS | UMPL | **PROMOTION** |
| Transolver | 10% | 2.49e-1 | 1.90e-1 | 1.85e-1 | 1.72e-1 | 1.56e-1 | 1.41e-1 | **10.16%** |
| | 30% | 2.49e-1 | 1.54e-1 | 1.44e-1 | 1.43e-1 | 1.21e-1 | 1.05e-1 | **13.22%** |
| AeroGTO | 10% | 1.77e-1 | 9.30e-2 | 7.20e-2 | 6.97e-2 | 6.24e-2 | 5.25e-2 | **15.19%** |
| | 30% | 1.77e-1 | 7.64e-2 | 6.46e-2 | 5.91e-2 | 4.74e-2 | 4.13e-2 | **12.47%** |

the PL method exhibits clear overfitting in the early training stages, UMPL avoids this issue, indicating superior generalization in the presence of distribution shifts.

Furthermore, Figure 11, Figure 12, Figure 13 and Figure 14 visualize performance comparisons and corresponding uncertainty estimations between UMPL and baseline methods on the Darcy Flow, NSM2d, Black Sea, and Plasma ICP datasets, respectively. These results demonstrate that UMPL not only achieves lower generalization errors but also provides reliable uncertainty estimates.

Notably, we also compare UMPL with heteroscedastic regression models trained solely on labeled data, using AeroGTO as the base surrogate model. As shown in Figure 15 (Black Sea dataset) and Figure 16 (NSM2d dataset), UMPL delivers more accurate and calibrated uncertainty estimates on unseen test data. In contrast, supervised-only models tend to produce overconfident but erroneous predictions.

### F.3 Quantitative Validation of Uncertainty-Error Correlation

To quantitatively validate that the predicted uncertainty reliably correlates with the actual error magnitude, we conduct comparative experiments on the AeroGTO framework using the Darcy

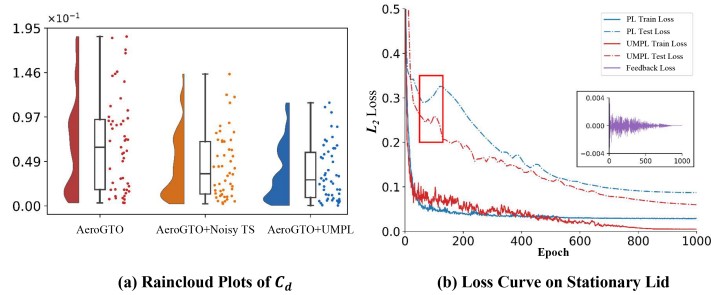

(a) **Raincloud Plots of $C_d$**

(b) **Loss Curve on Stationary Lid**

Figure 10: (a) Raincloud plot for $C_d$ prediction in Ahmed dataset. (b) Training and testing loss curve on the Stationary Lid dataset.

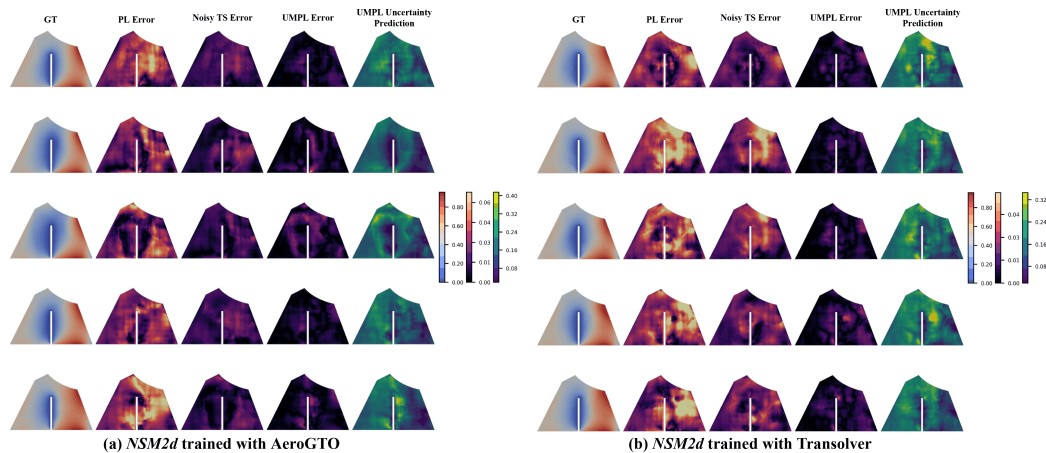

(a) *NSM2d* **trained with AeroGTO**

(b) *NSM2d* **trained with Transolver**

Figure 11: Comparison of our method against baseline methods on the Darcy Flow dataset and uncertainty estimation of our method.

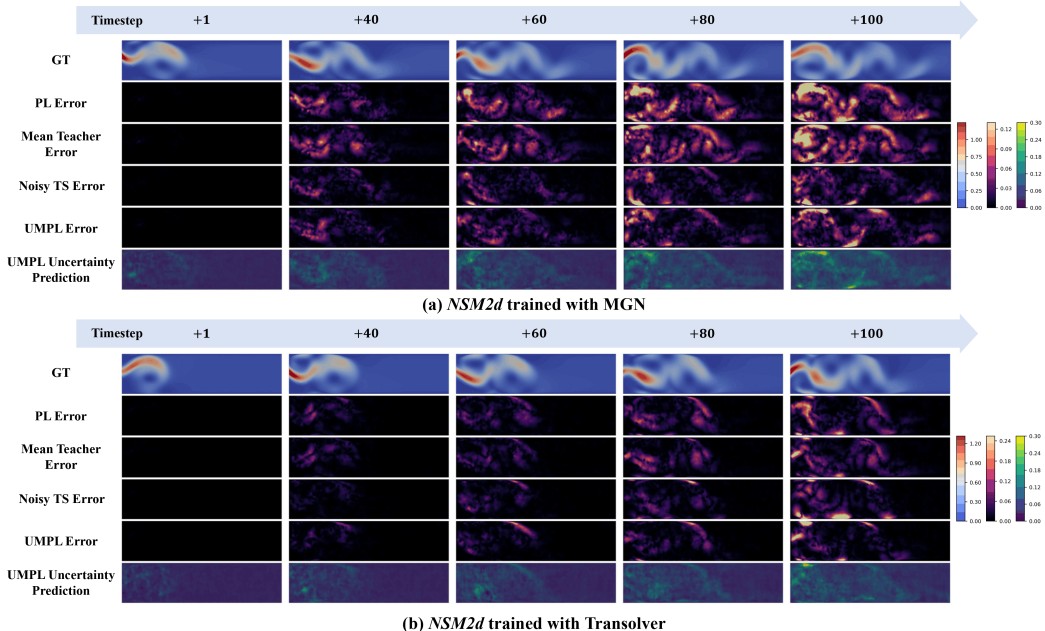

(a) *NSM2d* **trained with MGN**

(b) *NSM2d* **trained with Transolver**

Figure 12: Comparison of our method against baseline methods on the NSM2d dataset and uncertainty estimation of our method.

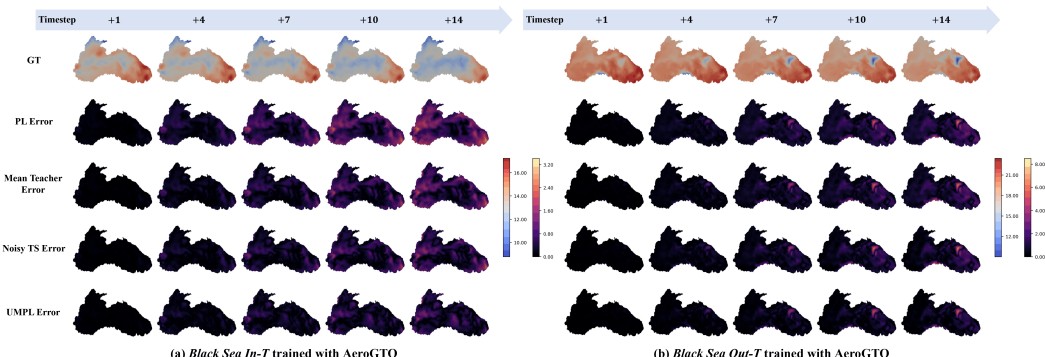

Figure 13: Comparison of our method against baseline methods on the Black Sea dataset.

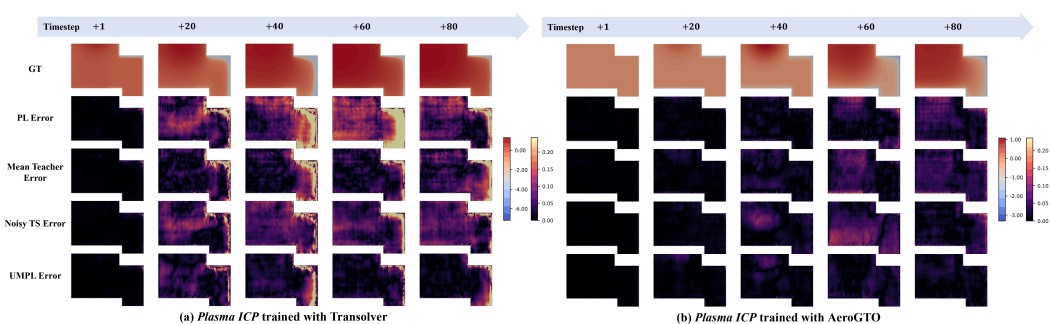

Figure 14: Comparison of our method against baseline methods on the Plasma ICP dataset.

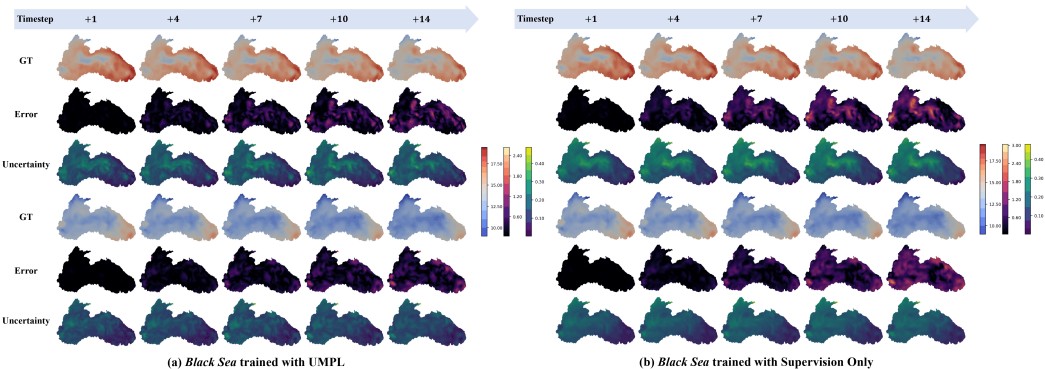

Figure 15: Prediction errors and uncertainty estimates of our method compared to the Supervision Only method on the Black Sea test dataset.

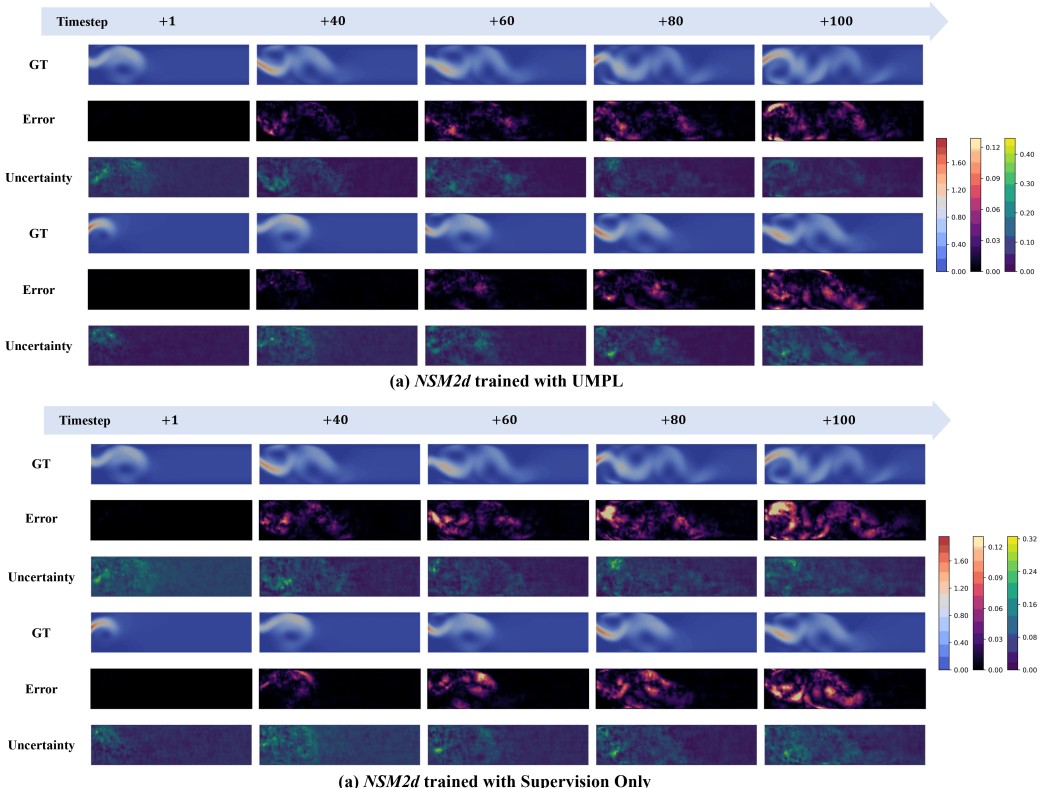

(a) *NSM2d* trained with UMPL

(a) *NSM2d* trained with Supervision Only

Figure 16: Prediction errors and uncertainty estimates of our method compared to the Supervision Only method on the NSM2d test dataset.

Flow and NSM2d datasets. The evaluation considers both accuracy and computational efficiency, employing miscalibration area (MA) and root mean squared calibration error (RMSCE) as calibration metrics, and $L_2$ error as the prediction accuracy metric.

Table 10 summarizes the comparison among different uncertainty estimation approaches, including Monte Carlo Dropout, SVGD [40], feature-WGD [65], and ensemble-based variants. Our proposed UMPL achieves the lowest MA and RMSCE with competitive training efficiency, demonstrating its ability to generate well-calibrated and reliable uncertainty estimates.

Table 10: Comparison of uncertainty estimation methods on the Darcy Flow dataset using AeroGTO as the base forecaster.

| Method | MA | RMSCE | $L_2$ | Training Time (h) | Memory (M) |
|---|---|---|---|---|---|
| EMC Dropout (Teacher) | 0.2123 | 0.2492 | 0.0388 | **0.52** | 2302 |
| MC Dropout | 0.1781 | 0.2333 | 0.0376 | 3.31 | 16249 |
| SVGD [40] | 0.1736 | 0.1954 | 0.0390 | 9.98 | **2253** |
| feature-WGD [65] | 0.1654 | 0.1880 | 0.0322 | 3.50 | 2567 |
| Latent-UQ (Student) | 0.1561 | 0.1778 | 0.0382 | 0.96 | 3243 |
| Ensemble [30] | 0.1847 | 0.2044 | 0.0319 | 9.68 | 2395 |
| Ensemble Latent-UQ [63] | 0.1297 | 0.1455 | 0.0294 | 13.39 | 3887 |
| **UMPL** | **0.0891** | **0.1011** | **0.0256** | 2.21 | 3257 |

As shown in Table 11, UMPL consistently achieves superior calibration and predictive accuracy compared with MC Dropout and Latent-UQ, indicating its robustness across different flow configurations.

These results highlight UMPL's strong capability to maintain a favorable balance between accuracy and efficiency while achieving reliable uncertainty calibration. Moreover, both the teacher and

Table 11: Comparison of uncertainty estimation methods on the NSM2d dataset.

| Method | MA | RMSCE | $L_2$ |
|---|---|---|---|
| Latent-UQ (Student) | 0.1959 | 0.2404 | 0.1728 |
| MC Dropout | 0.2841 | 0.3183 | 0.1734 |
| **UMPL** | **0.1127** | **0.1393** | **0.1213** |

student components in UMPL can be replaced by more advanced architectures to further enhance performance, albeit at the cost of higher computational complexity.

In addition, we replaced UMPL's uncertainty components with more advanced estimators, as shown in Table 12. While these alternatives achieved better calibration, they brought only marginal improvements in predictive accuracy and incurred significantly higher computational cost.

Table 12: Calibration and efficiency comparison with separate Teacher (T) and Student (S) components.

| Teacher (T) | Student (S) | MA | RMSCE | L2 | Training Time (h) | Memory (M) |
|---|---|---|---|---|---|---|
| EMC Dropout | Latent-UQ | 0.0891 | 0.1011 | 0.0256 | 2.21 | 3257 |
| SVGD | Latent-UQ | 0.0873 | 0.0983 | 0.0248 | 20.42 | 3677 |
| feature-WGD | Latent-UQ | 0.0842 | 0.0951 | 0.0243 | 6.53 | 5597 |
| feature-WGD | Ensemble Latent-UQ | 0.0773 | 0.0877 | 0.0239 | 28.43 | 6273 |

## F.4 Performance of UMPL under Non-Gaussian Uncertainty

To investigate how UMPL handles physical systems characterized by asymmetric or heavy-tailed uncertainty, we designed a one-dimensional Burgers-type PDE with a small diffusion coefficient as a testbed and adopted DeepONet as the backbone model. This setting introduces shocks and discontinuities, resulting in a highly non-Gaussian regime where models assuming Gaussian uncertainty often degrade. The governing equation is given by:

$$
\begin{cases}
\dfrac{\partial u}{\partial t} - \mu \dfrac{\partial^2 u}{\partial x^2} + u \dfrac{\partial u}{\partial x} + \epsilon u^3 = 0, \\
u(x,0) = ax^3 - (a+1)x, \quad x \in [-1,1], \\
-\mathbf{n} \cdot \dfrac{\partial u}{\partial x}\Big|_{x=\pm 1} = 0.
\end{cases}
\tag{62}
$$

Table 13 reports the quantitative comparison. The baseline UMPL, which assumes Gaussian uncertainty, exhibits degraded calibration performance under this non-Gaussian condition. To mitigate this issue, we replaced the teacher's epistemic uncertainty model with feature-WGD [65], which better captures heavy-tailed priors, and substituted the student's aleatoric uncertainty model with a Mixture Density Network (MDN) [55], capable of modeling multimodal and skewed distributions. This enhanced version substantially improves both calibration and predictive accuracy at the cost of increased computation.

Table 13: Results on a one-dimensional Burgers equation under non-Gaussian uncertainty using DeepONet as the backbone. "/" indicates no uncertainty estimate.

| Method | MA | RMSCE | L2 | Training Time (h) | Memory (M) |
|---|---|---|---|---|---|
| Supervised | / | / | 0.0499 | 1.42 | 1261 |
| UMPL | 0.3004 | 0.3448 | 0.0528 | 1.62 | 1439 |
| Improved UMPL (feature-WGD + MDN) | 0.1408 | 0.1590 | 0.0432 | 2.61 | 1789 |

These results indicate that assuming Gaussian uncertainty limits model calibration and predictive fidelity in non-Gaussian regimes. By integrating a more expressive epistemic uncertainty model

(feature-WGD) and a heavy-tailed noise model (MDN), UMPL achieves substantially better robustness and accuracy with moderate computational overhead. Overall, the framework demonstrates a flexible, modular design that enables an effective trade-off between efficiency and predictive reliability across diverse uncertainty structures.

## F.5 Extended Ablation Analysis

### F.5.1 Sensitivity and Robustness Analysis

We evaluate the sensitivity and robustness of UMPL on the out-of-distribution Stationary Lid-Driven Cavity dataset using AeroGTO as the backbone. We first probe robustness to aleatoric perturbations by injecting Gaussian noise of varying magnitude into the student's predicted uncertainty during training. As summarized in Table 14, uncertainty calibration (MA, RMSCE) degrades steadily with increasing noise, whereas predictive accuracy ($L_2$) remains stable under small perturbations but drops sharply with large noise. Overall, UMPL is robust to moderate noise yet sensitive to severe distortion.

Table 14: Sensitivity to injected Gaussian noise in the student's predicted uncertainty on the OOD Stationary Lid dataset.

| Noise Magnitude ($\sigma$) | MA | RMSCE | $L_2$ |
|---|---|---|---|
| 0.05 | 0.0703 | 0.0828 | 0.0634 |
| 0.10 | 0.0717 | 0.0879 | 0.0634 |
| 0.20 | 0.0779 | 0.0942 | 0.0636 |
| 0.50 | 0.0908 | 0.1053 | 0.0663 |
| 1.00 | 0.2156 | 0.2655 | 0.1499 |

We then study two design hyperparameters under the same setting: the PUF threshold (which filters pseudo-labels) and the feedback strength (which weights uncertainty-guided correction). Results in Tables 15–16 show that a lower PUF threshold can better filter noisy labels on harder datasets, though its effect is mild here. For feedback strength, too small values suppress pseudo-label correction (reducing UMPL toward plain pseudo-labeling), while overly large values inject noise and degrade label quality and student learning.

Table 15: Sensitivity to the PUF threshold on the OOD Stationary Lid dataset.

| PUF Threshold | MA | RMSCE | $L_2$ |
|---|---|---|---|
| 100% | 0.0679 | 0.0802 | 0.0633 |
| 95% | 0.0661 | 0.0783 | 0.0632 |
| 90% | 0.0707 | 0.0809 | 0.0638 |

Table 16: Sensitivity to feedback strength on the OOD Stationary Lid dataset.

| Uncertainty Strength | MA | RMSCE | $L_2$ |
|---|---|---|---|
| $\times 0.1$ | 0.0711 | 0.0838 | 0.0752 |
| $\times 1$ | 0.0679 | 0.0802 | 0.0633 |
| $\times 5$ | 0.0825 | 0.0974 | 0.0882 |
| $\times 10$ | 0.1273 | 0.1503 | 0.1367 |

Beyond these factors, we further investigate three UMPL hyperparameters on the same dataset: (i) the EMC update coefficient $\alpha$ in the teacher, (ii) the dropout rate used for Monte Carlo estimation, and (iii) the initial top-$k$ selection ratio for pseudo-label masking. In each experiment, one parameter is varied while others are held at their respective optima. The results in Figure 17 indicate non-monotonic trends for each parameter, highlighting UMPL's sensitivity to these choices and the importance of careful tuning. Exploring adaptive or data-driven strategies for hyperparameter selection is left for future work.

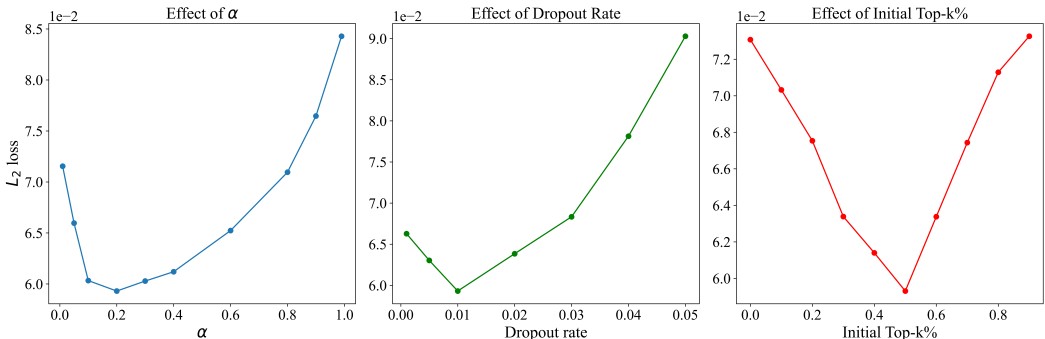

Figure 17: Ablation results for key UMPL hyperparameters on the Stationary Lid dataset: EMC update coefficient $\alpha$, MC-dropout rate $p$, and initial top-$k$ ratio for pseudo-label masking. Each curve varies one parameter while fixing others at their optimal values.

### F.5.2  Noise Robustness Analysis

While our main benchmarks do not explicitly include noisy observations and the current teacher primarily captures epistemic uncertainty, we further study noise robustness on Darcy Flow (AeroGTO backbone) by injecting input-dependent noise at varying levels. Results in Table 17 show performance degradation as noise increases. To improve robustness under noisy observations, we augment the teacher to incorporate both epistemic and aleatoric uncertainty, following [25], and introduce an ensemble student, which enhances predictive performance in the noisy regime.

Table 17: Darcy Flow with input-dependent noise: calibration (MA, RMSCE) and accuracy ($L_2$).

| Noise Level | MA | RMSCE | $L_2$ |
|---|---|---|---|
| 5% | 0.0996 | 0.1117 | 0.0257 |
| 10% | 0.1076 | 0.1207 | 0.0264 |
| 20% | 0.1286 | 0.1443 | 0.0302 |
| 40% | 0.1755 | 0.1975 | 0.0386 |
| 40%, Improved UMPL | 0.1335 | 0.1498 | 0.0348 |

## G  Borader Impact

The UMPL framework significantly enhances surrogate modeling performance, particularly under out-of-distribution and distribution shift scenarios. This improved generalization is valuable for tasks such as fluid dynamics modeling, climate prediction, and molecular simulations, where models often face data distributions that differ from training conditions. By enabling uncertainty-aware learning with limited supervision, UMPL reduces reliance on large labeled datasets and improves model robustness. In future work, we plan to extend UMPL to real-world scenarios and more complex applications, bridging the gap between research and deployment in science and engineering domains.

