# OpenReview forum: "Uncertainty-Informed Meta Pseudo Labeling for Surrogate Modeling with Limited Labeled Data"
_NeurIPS.cc/2025/Conference — NeurIPS 2025 poster_

### Official Review · Reviewer_FbuM · 2025-07-02

**Clarity:** 3
**Significance:** 3
**Originality:** 2
**Rating:** 4
**Confidence:** 3

**Summary:**

This paper proposes Uncertainty-Informed Meta Pseudo Labeling, a semi-supervised learning framework tailored for surrogate modeling of physical systems governed by PDEs under limited labeled data. The framework builds upon a teacher-student architecture, where the teacher estimates epistemic uncertainty via EMC Dropout to generate pseudo-labels, and the student learns from these pseudo-labels while modeling aleatoric uncertainty via heteroscedastic regression. A key contribution is the use of uncertainty-informed feedback: the student’s uncertainty guides refinement of the teacher’s pseudo-labels, enabling a closed-loop meta-learning process. The method improves generalization under distribution shifts across multiple tasks.

**Questions:**

listed in the weakness

**Ethical Concerns:**

["NO or VERY MINOR ethics concerns only"]

**Final Justification:**

The author has solved my concern, which is the performance of the uncertainty estimation module.

**Limitations:**

Yes. The paper includes a discussion on limitations in the appendix and acknowledges challenges such as computational cost and the assumption of reliable uncertainty estimation.

**Quality:**

3

**Strengths And Weaknesses:**

Strengths:
1. Integration of dual uncertainty modeling in a meta-learning loop.
2. Efficient and theoretically grounded uncertainty estimation.
3. Strong empirical results on PDEs and real-world datasets.


Weaknesses:
1. The key issue here is the contribution. Although this work proposed multiple ways to improve uncertainty estimation in terms of reliability and efficiency. However, the core part of this work is the SSL, which just use the uncertainty as a tool. I think the best way to justify the contribution should focus on SSL instead of the accuract uncertainty.
2. It is good to achieve a better uncertainty estimation, but why don't just use most up to date uncertainty estimation methods or  calibration methods that achieve better performance on universal accepted benchmarks?
3. The proposed uncertainty estimation method is not well evaluated on widely-accepted benchmarks, if the author believe the proposed method is better than other uncertainty estimation methods, you can include more experiments to justify it.
4. The computational overhead analysis is lacking.
5. How sensitive is the method to the quality of aleatoric uncertainty estimates from the student? The robustness of the method to different PUF thresholds or uncertainty injection strengths need to be better discussed.

---

> ### Author Rebuttal · Authors · 2025-07-30
>
> Thank you for your constructive comments. We address each point below.
>
> >Q1: The core part of this work is the SSL, which just use the uncertainty as a tool.
>
> **A1:** We thank the reviewer for the valuable comment. We agree that our work falls within the scope of semi-supervised learning (SSL), where uncertainty is employed to enhance pseudo-label quality. **The central objective of UMPL is to improve model generalization to unseen data by introducing a role-aware dual uncertainty framework.** This framework leverages epistemic uncertainty for teacher selection and aleatoric uncertainty for student feedback, forming a closed-loop refinement mechanism. Through this design, UMPL effectively exploits unlabeled data to enhance prediction accuracy in low-label scientific modeling settings. We will revise the manuscript to better highlight this focus.
>
> >Q2: Why don't just use most up to date uncertainty estimation methods or calibration methods?
>
> **A2:** We appreciate the reviewer’s suggestion. **Our main goal is to improve model accuracy in low-label settings by leveraging unlabeled data.** While more advanced uncertainty estimators exist, they offer limited accuracy gains at significantly higher computational cost. UMPL adopts a balanced choice based on this trade-off, and already outperforms baselines (shown in Q3&A3). Moreover, **UMPL is modular and can flexibly integrate more advanced teacher/student models if needed.** We will clarify this design choice and comparative results in the revised version.
>
> >Q3: The proposed uncertainty estimation method is not well evaluated on widely-accepted benchmarks.
>
> **A3:** We thank the reviewer for the helpful suggestion. Systematic comparisons of uncertainty estimation in PDE-based modeling are scarce—[1] is the only work that benchmarks multiple methods on the LE-PDE network to our best knowledge. To validate our approach, we conducted additional experiments on the public Darcy Flow dataset using AeroGTO, comparing UMPL with several uncertainty modeling baselines.
>
> Method|MA|RMSCE|L2|Training Time(h)|Memory(M)
> -|-|-|-|-|-
> EMC Dropout(Teacher)|0.2123|0.2492|0.0388|0.52|2302
> MC Dropout|0.1781|0.2333|0.0376|3.31|16249
> SVGD [2]|0.1736|0.1954|0.0390|9.98|2253
> feature-WGD [3]|0.1654|0.1880|0.0322|3.50|2567
> Latent-uq(Student)|0.1561|0.1778|0.0382|0.96|3243
> Ensemble [4]|0.1847|0.2044|0.0319|9.68|2395
> Ensemble latent-uq [1]|0.1297|0.1455|0.0294|13.39|3887|
> UMPL|0.0891|0.1011|0.0256|2.21|3257
>
> We use miscalibration area (MA) and root mean squared calibration error (RMSCE) as metrics to evaluate uncertainty quality. Results show that UMPL consistently achieves better uncertainty quality and predictive accuracy. We will include these results and clarify benchmarking choices in the revision. In addition, we replaced UMPL’s uncertainty components with more advanced estimators. **While these alternatives achieved better calibration, they brought only marginal improvements in predictive accuracy and incurred significantly higher computational cost.**
>
> Method|MA|RMSCE|L2|Training Time(h)|Memory(M)
> -|-|-|-|-|-
> T(EMC Dropout)+S(Latent-uq)|0.0891|0.1011|0.0256|2.21|3257
> T(SVGD)+S(Latent-uq)|0.0873|0.0983|0.0248|20.42|3677
> T(feature-WGD)+S(Latent-uq)|0.0842|0.0951|0.0243|6.53|5597
> T(feature-WGD)+S(Ensemble latent-uq)|0.0773|0.0877|0.0239|28.43|6273
>
> >Q4: The computational overhead analysis is lacking.
>
> **A4:** Thanks for the question. All experiments were conducted on a single NVIDIA RTX 4090 GPU. We report results on AeroGTO model across three representative tasks—2D steady-state, 2D transient, and 3D simulation—as summarized below.
>
> Dataset|Darcy (bs=4)||NSM2d (bs=4)||Ahmed (bs=1)||
> -|-|-|-|-|-|-
> Method|Memory(M)|Training Time(h)|Memory(M)|Training Time(h)|Memory(M)|Training Time(h)
> Supervised|2297|0.45|9727|3.75|9375|3.53
> Pseudo Label|3843|1.08|16567|9.04|15947|8.73
> Mean Teacher|3843|1.21|16567|10.17|15947|9.57
> Noisy TS|2304|1.77|9813|13.21|9477|12.48
> Supervised+uq|3243|0.96|13578|7.69|12986|7.36
> UMPL|3257|2.21|13792|16.48|13290|15.56
>
> Here, Supervised+uq refers to the student model used in UMPL. As shown, **UMPL introduces almost no additional memory overhead**, thanks to the REINFORCE-style update that avoids the costly backpropagation chain from the student to the teacher. This significantly reduces computational and memory complexity, **at the cost of increased training time**.
>
> >Q5: The sensitivity and robustness analysis.
>
> **A5:** Thank you for the helpful suggestion. We evaluated sensitivity to aleatoric uncertainty on the OOD Stationary Lid-Driven Cavity dataset using AeroGTO as the backbone. Gaussian noise of varying magnitudes was added to the student’s predicted uncertainty during training.
>
> Noisy Magnitude($\sigma$)|MA|RMSCE|L2
> -|-|-|-|
> 0.05|0.0703|0.0828|0.0634
> 0.1|0.0717|0.0879|0.0634
> 0.2|0.0779|0.0942|0.0636
> 0.5|0.0908|0.1053|0.0663
> 1.0|0.2156|0.2655|0.1499
>
> The result shows that while uncertainty calibration degrades steadily with noise, the predictive accuracy remains stable under small perturbations. However, large noise leads to sharp performance drops, indicating that **UMPL is robust to moderate noise but sensitive to severe distortion.**
>
> We also evaluated the sensitivity to the PUF threshold and feedback strength under the same setting:
>
> PUF threshold|MA|RMSCE|L2
> -|-|-|-|
> 100%|0.0679|0.0802|0.0633
> 95%|0.0661|0.0783|0.0632
> 90%|0.0707|0.0809|0.0638
>
> Uncertainty strength|MA|RMSCE|L2
> -|-|-|-|
> x0.1|0.0711|0.0838|0.0752
> x1|0.0679|0.0802|0.0633
> x5|0.0825|0.0974|0.0882
> x10|0.1273|0.1503|0.1367
>
> **A lower PUF helps filter noisy labels on harder datasets, though its impact is mild on this dataset. For feedback strength, too small values lead to no pseudo-label correction (reducing UMPL to plain PL), while overly large values inject noise, degrading label quality and student learning.**
>
> Reference
>
> [1] Wu, Tailin, et al. "Uncertainty quantification for forward and inverse problems of pdes via latent global evolution." Proceedings of the AAAI Conference on Artificial Intelligence. Vol. 38. No. 1. 2024.
>
> [2] Liu, Q. and Wang, D., 2016. Stein variational gradient descent: A general purpose bayesian inference algorithm. Advances in neural information processing systems, 29.
>
> [3] Yashima, Shingo, et al. "Feature space particle inference for neural network ensembles." International Conference on Machine Learning. PMLR, 2022.
>
> [4] Lakshminarayanan, Balaji, Alexander Pritzel, and Charles Blundell. "Simple and scalable predictive uncertainty estimation using deep ensembles." Advances in neural information processing systems 30 (2017).

---

### Official Review · Reviewer_9hkT · 2025-07-03

**Clarity:** 3
**Significance:** 3
**Originality:** 3
**Rating:** 5
**Confidence:** 3

**Summary:**

The paper proposes a semi-supervised teacher-student framework for emulators with limited simulation data. The teacher generates pseudo-labels using MC-Dropout to predict epistemic uncertainty, while the student learns from these labels while predicting aleatoric uncertainty via variance prediction. The student's uncertainty provides feedback to refine the teacher's pseudo-labels in a meta-learning loop and adds spatial coherence. Evaluation of the system is presented from scientific tasks and distribution shifts.

**Questions:**

1. Both epistemic (MC-Dropout) and aleatoric (heteroscedastic variance) uncertainty predictors assume symmetric Gaussian distributions. How does UMPL handle physical systems with asymmetric or heavy-tailed uncertainties (e.g., shock waves, bifurcation points) where these assumptions fail? Did you evaluate on such systems?
2. Why are standard uncertainty calibration metrics absent? Can you show quantitative proof that the predicted uncertainty reliably correlates with error magnitude (not just direction) in OOD regions?"
3. Could you report computing resources?

**Ethical Concerns:**

["NO or VERY MINOR ethics concerns only"]

**Final Justification:**

The authors have addressed most of my questions. I am increasing my score at 5.

**Limitations:**

yes

**Quality:**

3

**Strengths And Weaknesses:**

**Strengths**
* Original feedback closed loop between teacher-epistemic and student-aleatoric
* The mesh-aware spatial coherence for the pseudo-labels
* Extensive experimental section on various architectures

**Weaknesses**
* Questionable uncertainty method: the paper relies on MC-Dropout to estimate the teacher-epistemic, known to struggle when the true uncertainty isn't nicely behaved and bell-shaped (Gaussian) – common in messy real-world physics like shocks or abrupt changes.
* Reliable uncertainties not shown: there should be some experimental proof that the uncertainty prediction are trustworthy using checks like calibration metrics or reliability diagrams. Aren' t the errors remain high in complex tasks like NSM2D?
* Potentially overly easy experiments: benchmarks used clean, smooth simulation data without the real-world noise or sharp changes one could encounter. This likely explains why even the basic "Pseudo Label" baseline beat supervised learning – a result that often doesn't hold up with noisier or more complex data.  I am not convinced that the method would hold against the hard distribution shifts or more complex noise common in real data

---

> ### Author Rebuttal · Authors · 2025-07-30
>
> Thank you for the positive feedback.
>
> >Q1: Quantitative proof that the predicted uncertainty reliably correlates with error magnitude.
>
> **A1:** Thank you for the insightful question. We have added experiments comparing our method with other uncertainty estimation approaches on AeroGTO for the Darcy Flow dataset in terms of both accuracy and efficiency, using miscalibration area (MA) and root mean squared calibration error (RMSCE) as evaluation metrics:
>
> Method|MA|RMSCE|L2|Training Time(h)|Memory(M)
> -|-|-|-|-|-
> EMC Dropout(Teacher)|0.2123|0.2492|0.0388|0.52|2302
> MC Dropout|0.1781|0.2333|0.0376|3.31|16249
> SVGD [1]|0.1736|0.1954|0.0390|9.98|2253
> feature-WGD [2]|0.1654|0.1880|0.0322|3.50|2567
> Latent-uq(Student)|0.1561|0.1778|0.0382|0.96|3243
> Ensemble [3]|0.1847|0.2044|0.0319|9.68|2395
> Ensemble latent-uq [4]|0.1297|0.1455|0.0294|13.39|3887|
> UMPL|0.0891|0.1011|0.0256|2.21|3257
>
> Additionally, we conducted a comparison on the NSM2d experiment:
>
> Method|MA|RMSCE|L2
> -|-|-|-
> Latent-uq(Student)|0.1959|0.2404|0.1728
> MC Dropout|0.2841|0.3183|0.1734
> UMPL|0.1127|0.1393|0.1213
>
> These comparisons highlight **UMPL's strong performance over baselines with a good balance between accuracy and efficiency.** The teacher and student components can be flexibly replaced with more advanced methods for further gains, albeit at higher computational cost.
>
> >Q2: The method may not hold up with the noise or real-world more complex data.
>
> **A2:** Thank you for the thoughtful comment. We have considered challenging scenarios in our experiments. For example, the Black Sea dataset is based on real ocean measurements with strong seasonal patterns and complex temporal evolution. It poses a difficult temporal OOD generalization problem, yet UMPL still achieves performance gains (albeit smaller than on other datasets). The Plasma ICP dataset involves distribution shift from low- to high-fidelity simulations. UMPL outperforms baselines using only 10% of high-fidelity data, showing strong data efficiency under realistic conditions.
>
> We acknowledge that our benchmarks do not include noisy data. The current teacher model focuses on epistemic uncertainty and does not explicitly model data noise. To address this, we conducted an additional experiment on the Darcy Flow dataset using the AeroGTO backbone, introducing varying levels of input-dependent noise.
>
> Noise|MA|RMSCE|L2
> -|-|-|-
> 5%|0.0996|0.1117|0.0257
> 10%|0.1076|0.1207|0.0264
> 20%|0.1286|0.1443|0.0302
> 40%|0.1755|0.1975|0.0386
> 40%,Improved UMPL|0.1335|0.1498|0.0348
>
> Results show that model performance degrades with added noise. **To improve robustness with noise, we incorporated both epistemic and aleatoric uncertainty in the teacher** following [5], and added an ensemble student model. This enhanced predictive performance under noisy conditions, which we will detail further in the revised version.
>
> >Q3: How does UMPL handle physical systems with asymmetric or heavy-tailed uncertainties?
>
> **A3:** Thank you for the insightful question. Our teacher and student models indeed assume Gaussian scenarios, which is suitable for most standard cases. However, we acknowledge that performance may degrade under shocks or discontinuities. To evaluate this, we designed an experiment using a 1D Burgers-type PDE with small diffusion coefficient:
>
> $\frac{\partial u}{\partial t}-\mu \frac{\partial ^2 u}{\partial x^2}+u \frac{\partial u}{\partial x}+\epsilon u^3=0,u_0(x)=ax^3-(a+1)x,x\in[-1,1],-\mathbf{n}\cdot\frac{\partial u}{\partial x}|_{x=-1,1}=0$
>
> Using DeepONet as the backbone, results show that the original model struggles under such non-Gaussian settings:
>
> Method|MA|RMSCE|L2|Training Time(h)|Memory(M)
> -|-|-|-|-|-
> Supervised|/|/|0.0499|1.42|1261
> UMPL|0.3004|0.3448|0.0528|1.62|1439
> Improved UMPL|0.1408|0.1590|0.0432|2.61|1789
>
>  We improved performance by replacing the teacher's epistemic model with feature-WGD and the student's aleatoric model with MDN [6], which better capture heavy-tailed distributions. This comes at higher computational cost. **Our framework thus offers a flexible trade-off between efficiency and accuracy, with modular components adaptable to specific scenarios.** We will include these findings in the revised appendix.
>
> >Q4: Could you report computing resources?
>
> **A4:** Thanks for the question. All experiments were conducted on a single NVIDIA RTX 4090 GPU. We report results on AeroGTO model across three representative tasks—2D steady-state, 2D transient, and 3D simulation—as summarized below.
>
> Dataset|Darcy (bs=4)||NSM2d (bs=4)||Ahmed (bs=1)||
> -|-|-|-|-|-|-
> Method|Memory(M)|Training Time(h)|Memory(M)|Training Time(h)|Memory(M)|Training Time(h)
> Supervised|2297|0.45|9727|3.75|9375|3.53
> Pseudo Label|3843|1.08|16567|9.04|15947|8.73
> Mean Teacher|3843|1.21|16567|10.17|15947|9.57
> Noisy TS|2304|1.77|9813|13.21|9477|12.48
> Supervised+uq|3243|0.96|13578|7.69|12986|7.36
> UMPL|3257|2.21|13792|16.48|13290|15.56
>
> Here, Supervised+uq refers to the student model used in UMPL. As shown, **UMPL introduces almost no additional memory overhead**, thanks to the REINFORCE-style update that avoids the costly backpropagation chain from the student to the teacher. This significantly reduces computational and memory complexity, **at the cost of increased training time**.
>
> Reference
>
> [1] Liu, Q. and Wang, D., 2016. Stein variational gradient descent: A general purpose bayesian inference algorithm. Advances in neural information processing systems, 29.
>
> [2] Yashima, Shingo, et al. "Feature space particle inference for neural network ensembles." International Conference on Machine Learning. PMLR, 2022.
>
> [3] Lakshminarayanan, Balaji, Alexander Pritzel, and Charles Blundell. "Simple and scalable predictive uncertainty estimation using deep ensembles." Advances in neural information processing systems 30 (2017).
>
> [4] Wu, Tailin, et al. "Uncertainty quantification for forward and inverse problems of pdes via latent global evolution." Proceedings of the AAAI Conference on Artificial Intelligence. Vol. 38. No. 1. 2024.
>
> [5] Kendall, Alex, and Yarin Gal. "What uncertainties do we need in bayesian deep learning for computer vision?." Advances in neural information processing systems 30 (2017).
>
> [6] Thakur, Akshay, and Souvik Chakraborty. "MD-NOMAD: Mixture density nonlinear manifold decoder for emulating stochastic differential equations and uncertainty propagation." arXiv preprint arXiv:2404.15731 (2024).

---

> ### Author Response · Authors · 2025-08-06
> **Looking forward to your reply**
>
> Dear Reviewer 9hkT,
>
> We have provided new experiments and discussions according to your valuable suggestions, which have been absorbed into the revised manuscript. We hope that the new manuscript is made to be stronger with your suggestions.
>
> As the rebuttal deadline is approaching, we sincerely look forward to your reply. Thanks so much for your time and effort!
>
> Best Regards,
>
> Authors of Submission 10749

---

### Official Review · Reviewer_VXJf · 2025-07-03

**Clarity:** 3
**Significance:** 3
**Originality:** 2
**Rating:** 4
**Confidence:** 2

**Summary:**

This paper tackles the challenge of building robust surrogate models for complex physical systems, especially under conditions of limited labeled data, distribution shifts or OOD scenarios. Traditional DNNs and neural operators require large labeled datasets, which are often costly or impractical to obtain in scientific domains but the problem arises as, even though semi-supervised learning offers a way to leverage abundant unlabeled data, it suffers from noisy pseudo-labels that can degrade model performance.

To tackle this the authors propose Uncertainty-Informed Meta Pseudo Labeling (UMPL), a semi-supervised learning framework that improves pseudo-label quality by incorporating uncertainty estimates in a setup of a teacher-student meta-learning framework where they have roles based on uncertainty. Specifically, the teacher model generates pseudo-labels along with epistemic uncertainty, which reflects the model’s confidence and helps identify uncertain predictions under distribution shifts. The student model however learns from these pseudo-labels in order to estimate the aleatoric uncertainty, capturing inherent data noise and label uncertainty. This separation of uncertainty types aligns with the distinct roles of teacher and student and leads to more robust and interpretable uncertainty estimates. The student provides feedback to the teacher based on aleatoric uncertainty, guiding the teacher to refine pseudo-labels iteratively. This meta-learning loop enhances pseudo-label quality and model generalization, especially in uncertain or underrepresented regions of the data.

**Questions:**

As mentioned above:

As I am unfamiliar with the setting of modeling high-dimensional PDE systems I struggle to find the motivation for the choice of relative L_2 error compared to just the standard L2 loss. Can the authors explain this motivation?

How long did it take to run the experiments and did you use GPUs or CPUs? Could you please elaborate more on why do you not report statistical significance? Can you please include significance for at least a couple of experiments in Table 1 or 2?

Typo in Appendix H title: "experiential settings"

**Ethical Concerns:**

["NO or VERY MINOR ethics concerns only"]

**Final Justification:**

I am still hesitant and reluctant due to the lack of statistical significance of the results. The method seems to be extremely more computational heavy compared to the other methods both in terms of memory and training time. I am especially reluctant as the authors said they have not included statistical significance due to space constraints which seems odd.

However, as I am not well versed in the field and the after reading the other reviewers' comments, I have chosen to update my score from 3 to a 4. However, I still have some concerns and worries so will not give a larger score.

**Limitations:**

Yes

**Paper Formatting Concerns:**

No formatting concerns

**Quality:**

3

**Strengths And Weaknesses:**

The paper is well written and the motivation is clear. Although I am not that familiar with the practical problems and frameworks that solve (or at least tackle how to solve) problems like pseudo-labeling of high-dimensional PDE systems, it seems that the problem is well formulated and the authors motivate it well.

Regarding the contribution of the paper, in their second bullet point the authors claim that the role-aware design leads to more robust and interpretable uncertainty estimates, however without backing it up with any claims. It would be good if the authors could point out or summarize the sections, results (figures/tables) which back this claim up. At the moment it seems a bit unjustified. The same comment stands for the third bullet point, in which the authors claim that the targeted feedback significantly improves the quality of pseudo labels across iterations without claiming why that is the case.

As I am unfamiliar with the setting of modeling high-dimensional PDE systems I struggle to find the motivation for the choice of relative L_2 error compared to just the standard L2 loss. Can the authors explain the motivation why the loss choice in (1)? Also, why have the authors chosen an encoder embedding? Is this a popular model choice in the student-teacher literature? Is there a reason why one would expect this would work better for modeling PDE systems? Both fall a bit flat unmotivated.

The same comment stands for their choice of MC Dropout. Although the simplest to implement and most straightforward to train from my experience with BNNs, the authors should argue more why they have chosen it, and not for example SVGD[1], fSVGD[2] or any other Bayesian NN choice, and I think that they should perform benchmarks/baselines to corroborate their choice. In Appendix C the authors give a nice Bayesian interpretation of MC Dropout which I am thankful for, but I was a bit disappointed that they did not deliberate anything about why specifically choosing it over other methods.

Regarding Figure 2. it is not really clear what the point of the figure is, the error between the proposed method and Noisy TS is differently distributed and with different magnitudes and its not clear to me from the plot which one is better.

Regarding using REINFORCE as the solution to minimize the loss in (8) and deriving its gradient, it would be nice if the authors add a few comments regarding the REINFORCE rule; are there any downsides to using it, is it the standard in other literature, what are its benefits and so on. I am not well versed in this literature and to me, although all the design choices that the authors have made towards proposing the UMPL framework, they fall a bit flat. In general, is there any reason why we would expect each of them to work better separately and together, how do they interplay and can the authors show that other choices for the building blocks of UMPL would've resulted in inferior performance? The ablation study in Table 3 was indeed nice and provided one small step towards this but giving at least one or two examples of what happens with other choices of uncertainty estimation (e.g., Bayesian NNs) would've been nice to include.

One argument towards this could be the superior results that the authors observe in Tables 1 and 2, but these also fall a bit short as I don't know if these results are statistically significant or not. In answers to question 6 of the Q&A the authors mention that they introduce experiment setting and details but I couldn't find how long did it take to run your experiments and whether the authors ran them on GPUs vs CPUs. Furthermore, it is mentioned that the statistical significance is not reported as in fluid dynamics modeling using a fixed random seed shows consistent performance. I believe this claim needs to be backed up and at least commented in the main paper or the appendix. If the results were computationally expensive and you have limited resources this is fine but then this should be commented on too. If you did mention this somewhere in the appendix I would be happy if you can point me to it, but if your experiments did not require extensive computational resources I think that statistical significance needs to be added both to Tables 1 and 2 as well as the L2 error in Table 3.

[1] Liu, Q. and Wang, D., 2016. Stein variational gradient descent: A general purpose bayesian inference algorithm. Advances in neural information processing systems, 29.
[2] Wang, Z., Ren, T., Zhu, J. and Zhang, B., 2019. Function space particle optimization for Bayesian neural networks. arXiv preprint arXiv:1902.09754.

---

> ### Author Rebuttal · Authors · 2025-07-30
>
> Thank you for your careful reading of the paper and your constructive comments. We address each point below.
>
> >Q1:Request to clarify and support the claimed benefits of role-aware uncertainty modeling and targeted feedback.
>
> **A1:** In response to the request for evidence supporting the claims of “role-aware design” and “targeted feedback,” we provide the following clarifications:
>
> Regarding the second bullet point: “Role-aware design leads to more robust and interpretable uncertainty estimates” — our framework uses role-specific uncertainty: the teacher models epistemic uncertainty for distribution shifts, while the student captures aleatoric uncertainty for input-dependent noise, forming a complementary structure. This design is supported by:
>
> - Theoretical justification:
> Section 3.1 provides the theoretical foundation, where Theorem 3.1 introduces EMC Dropout for efficient epistemic uncertainty estimation on the teacher side, Equation (5) formulates the aleatoric uncertainty optimization for the student, and Theorem 3.2 establishes an upper bound on student error under noisy pseudo labels.
>
> - Empirical evidence:
> Figure 2 shows the student’s feedback loss aligns well with true error, demonstrating strong spatial adaptivity. Figures 3(a) demonstrates that our method, applied to the Ahmed dataset, provides more accurate uncertainty estimates compared to the approach that uses only the student model trained on labeled data.
>
> Regarding the third bullet point: “Targeted feedback significantly improves the quality of pseudo labels”
> This is supported both theoretically and empirically:
>
> - Theoretical justification:
> Section 3.2 introduces a point-wise feedback mechanism guided by the student’s aleatoric uncertainty. The pseudo labels are perturbed (Eq. 6), and the teacher is updated using a first-order gradient approximation (Eq. 10–12), which avoids expensive meta-gradient computation.
>
> - Empirical evidence:
> Figure 2 shows that feedback signals correlate strongly with true label error, guiding the teacher to focus on high-error regions and improving pseudo-label quality over time. Tables 1 and 2 demonstrate that UMPL consistently outperforms methods without targeted feedback (e.g., Pseudo Label, Mean Teacher), and Table 3 shows a 42.6% performance drop when the feedback module is removed, confirming its critical role.
>
> We acknowledge that the current version lacks direct quantitative analysis of pseudo-label quality progression. In the revised version, we will include comparisons of pseudo-label error curves over iterations to more explicitly demonstrate how targeted feedback improves pseudo-label quality.
>
> >Q2:Why to choose relative L2 error?
>
> **A2:** Thank you for the question. We adopt relative L2 error due to its clear advantages in scientific modeling, particularly in the following aspects:
>
> - It is **well-suited for multi-scale systems or quantities with heterogeneous units**, where standard L2 loss tends to overweight large-magnitude regions and ignore small but important ones.
>
> - In **multi-physics problems** (e.g., Plasma ICP), where predicted variables differ by orders of magnitude, standard L2 loss leads to gradient imbalance. **Relative L2 provides scale-invariant optimization that treats all fields fairly.**
>
> - This choice is **widely adopted in neural operator literature** (e.g., FNO, Transolver, AeroGTO), both as a training loss and evaluation metric, due to its physical interpretability and robustness.
>
> Therefore, relative L2 error not only serves as an effective evaluation metric but also as a more appropriate training objective.
>
> >Q3:Why to choose an encoder embedding? Is this a popular choice in the student-teacher literature? Is it work better for modeling PDE systems?
>
> **A3:** Thank you for the question. The use of encoder embeddings is motivated by the following:
>
> - **Unified representation of heterogeneous PDE inputs:** PDE problems involve geometry, physical parameters, and boundary conditions. The encoder maps these multimodal inputs into a common latent space, enhancing generalization. For instance, in the Plasma ICP case, the encoder helps integrate irregular mesh geometry and spatial parameters into a consistent representation.
>
> - **Widely used across domains:** Teacher–student frameworks commonly use encoder embeddings in tasks such as image classification[1] and defect detection[2], enabling stable pseudo-labeling and alignment between models.
>
> - **Adaptation to PDE-specific challenges:** For datasets with non-uniform meshes or complex geometries (e.g., Darcy Flow, NSM2d), the encoder learns geometry-invariant representations to improve spatial robustness. This strategy is also used in operator models like DeepONet and AeroGTO to align heterogeneous inputs into a unified latent space.
>
> We will clarify this design choice in the revised version.
>
> >Q4: Figure 2 isn't clear which method is better.
>
> **A4:** Thank you for the valuable feedback. In the revised version, we will replace the last subplot in Figure 2 with a comparison of Kendall correlation between feedback loss and true pseudo-label error for UMPL and Noisy TS. **On 100 unlabeled samples, UMPL achieves much stronger correlation**: 59 samples exceed 0.8 (vs. 5 for Noisy), and only 2 fall below 0.6 (vs. 11 for Noisy). This highlights that UMPL’s aleatoric uncertainty-guided feedback more reliably captures pseudo-label error, leading to more stable and robust training.
>
> >Q5: Comments regarding the REINFORCE rule.
>
> **A5:** Thank you for the question. We adopt REINFORCE to estimate the gradient in Eq.(8), primarily to avoid the costly backpropagation path from student to teacher, thus **significantly reducing memory and computational overhead.** REINFORCE is a standard technique and has been widely used in meta pseudo-labeling methods in computer vision. Its main limitation is **increased time cost due to repeated forward and backward passes.** We will clarify this design choice and trade-off in the revised version.
>
> >Q6: Why not choose other uq methods?
>
> **A6:** Thank you for the insightful question. As noted, we chose EMC mainly for its computational efficiency. While more accurate uq methods exist, EMC offers a good balance between cost and performance. **Our main goal is to use unlabeled data to boost prediction accuracy on unseen data.** We have added experiments comparing different uq methods on AeroGTO for Darcy Flow dataset in terms of both accuracy and efficiency, using miscalibration area (MA) and root mean squared calibration error (RMSCE) as evaluation metrics:
>
> Method|MA|RMSCE|L2|Training Time(h)|Memory(M)
> -|-|-|-|-|-
> EMC Dropout(Teacher)|0.2123|0.2492|0.0388|0.52|2302
> MC Dropout|0.1781|0.2333|0.0376|3.31|16249
> SVGD|0.1736|0.1954|0.0390|9.98|2253
> feature-WGD [3]|0.1654|0.1880|0.0322|3.50|2567
> Latent-uq(Student)|0.1561|0.1778|0.0382|0.96|3243
> Ensemble [4]|0.1847|0.2044|0.0319|9.68|2395
> Ensemble latent-uq [5]|0.1297|0.1455|0.0294|13.39|3887|
> UMPL|0.0891|0.1011|0.0256|2.21|3257
>
> As shown in the results, the teacher performs worse than advanced Bayesian methods (e.g., SVGD, feature-WGD), and the student underperforms compared to Ensemble latent-uq. However, both are more efficient. As discussed in Q7&A7, **Advanced replacements improve uncertainty estimation with minimal accuracy gains, at a much higher computational cost. Our framework remains flexible, enabling such substitutions depending on practical trade-offs.**
>
> >Q7: Other choices for the building blocks of UMPL?
>
> **A7:** We replaced the student and teacher components using the same setup as in Q6&A6. The performance comparison is shown below:
>
> Method|MA|RMSCE|L2|Training Time(h)|Memory(M)
> -|-|-|-|-|-
> T(EMC Dropout)+S(latent-uq)|0.0891|0.1011|0.0256|2.21|3257
> T(SVGD)+S(Latent-uq)|0.0873|0.0983|0.0248|20.42|3677
> T(feature-WGD)+S(Latent-uq)|0.0842|0.0951|0.0243|6.53|5597
> T(feature-WGD)+S(Ensemble latent-uq)|0.0773|0.0877|0.0239|28.43|6273
>
> These results confirm that replacing components leads to better performance, at the cost of increased computational overhead.
>
> >Q8: Computational resources & statistical significance?
>
> **A8:** Thanks for the question. All experiments were conducted on a single NVIDIA RTX 4090 GPU. We report results on AeroGTO model across three representative tasks—2D steady-state, 2D transient, and 3D simulation—as summarized below.
>
> Dataset|Darcy (bs=4)||NSM2d (bs=4)||Ahmed (bs=1)||
> -|-|-|-|-|-|-
> Method|Memory(M)|Training Time(h)|Memory(M)|Training Time(h)|Memory(M)|Training Time(h)
> Supervised|2297|0.45|9727|3.75|9375|3.53
> Pseudo Label|3843|1.08|16567|9.04|15947|8.73
> Mean Teacher|3843|1.21|16567|10.17|15947|9.57
> Noisy TS|2304|1.77|9813|13.21|9477|12.48
> Supervised+uq|3243|0.96|13578|7.69|12986|7.36
> UMPL|3257|2.21|13792|16.48|13290|15.56
>
> Due to main text space constraints, we reported only the mean over 5 trials with different seeds. Statistical significance results will be included in the appendix of the revised version.
>
> >Q9: Typo in Appendix H title.
>
> **A9:** Fixed it.
>
> Reference
>
> [1] Wang, Zhenbin, et al. "Metateacher: Coordinating multi-model domain adaptation for medical image classification." Advances in Neural Information Processing Systems 35 (2022): 20823-20837.
>
> [2] Zhao, Sinong, et al. "Meta pseudo labels for anomaly detection via partially observed anomalies." Engineering Applications of Artificial Intelligence 126 (2023): 106955.
>
> [3] Yashima, Shingo, et al. "Feature space particle inference for neural network ensembles." International Conference on Machine Learning. PMLR, 2022.
>
> [4] Lakshminarayanan, Balaji, Alexander Pritzel, and Charles Blundell. "Simple and scalable predictive uncertainty estimation using deep ensembles." Advances in neural information processing systems 30 (2017).
>
> [5] Wu, Tailin, et al. "Uncertainty quantification for forward and inverse problems of pdes via latent global evolution." Proceedings of the AAAI Conference on Artificial Intelligence. Vol. 38. No. 1. 2024.

---

> > ### Comment · Reviewer_VXJf · 2025-08-03
> >
> > I thank the authors for their additional included experiments and explanations. I hope that they incorporate everything they said in the paper as, if this is the case, I think the paper will be stronger and will have sufficient weight to be a conference paper.

---

> > > ### Author Response · Authors · 2025-08-04
> > > **Appreciation for the constructive comments**
> > >
> > > We sincerely appreciate your helpful suggestions! Thanks for your time and effort!

---

### Official Review · Reviewer_fwWx · 2025-07-09

**Clarity:** 2
**Significance:** 3
**Originality:** 3
**Rating:** 5
**Confidence:** 3

**Summary:**

This paper tackles the challenge of building surrogate models for physics-governed systems when only a small, costly set of simulation labels is available alongside abundant unlabeled data, proposing a semi-supervised teacher–student framework in which a lightweight Exponential-Moving Monte-Carlo Dropout (EMC-Dropout) teacher predicts system states and provides epistemic uncertainty, while a student network learns from those pseudo-labels, estimates aleatoric uncertainty, and feeds it back to the teacher. Training is further stabilized by Progressive Uncertainty Filtering, which gradually discards the noisiest pseudo-labeled regions, and by Supervised Anchoring, which periodically re-grounds the model on true labels. Evaluated by plugging into four distinct neural operators—DeepONet, MeshGraphNet, Transolver, and AeroGTO—across seven CFD and physics benchmarks spanning partial differential equations, real-world datasets, and large-scale 3-D simulations, the method achieves roughly a 14 % error reduction with only 10 % labeled data and markedly improves out-of-distribution and distribution-shift generalization.

**Questions:**

1. Could you elaborate on the computational resources your method requires—e.g., hardware, run-time, and memory overhead—so readers can better gauge its practical cost?

2. Within the teacher–student framework, is there a risk of model collapse, where the teacher and student converge to similar outputs on unseen data yet still generate poor labels relative to the true target distribution? - How to protect against this?

**Ethical Concerns:**

["NO or VERY MINOR ethics concerns only"]

**Final Justification:**

Authors have addressed most of my questions. I will maintain my score at 5.

**Limitations:**

Yes — the limitations are discussed in Appendix Section K.

**Paper Formatting Concerns:**

There are no significant issues with the paper’s formatting.

**Quality:**

3

**Strengths And Weaknesses:**

**Strengths**

1. **Novel teacher–student strategy** – The framework introduces an innovative teacher–student loop that produces more reliable pseudo-labels while simultaneously training the surrogate model.

2. **Architecture-agnostic design** – Because the method is model-agnostic, it can be plugged into virtually any neural-operator or surrogate-model architecture with minimal modification.

3. **Thorough empirical validation** – The authors support their claims with extensive experiments across multiple datasets and architectures, demonstrating consistently strong performance.



**Weaknesses**

1. **Potential computational burden** – Running two networks, multiple Monte-Carlo dropout passes, and iterative feedback may be resource-intensive. A dedicated section detailing the hardware used, run-times, and memory footprint would clarify the true computational cost.

2. **Clarity and presentation** – Several aspects of the manuscript hinder readability:

   a. Key variables and symbols should be defined in the main text rather than deferred to the appendix.

   b. Algorithm 1 should be included in the main body to better understand the whole procedure.

   c. Although Figure 1 is helpful in principle, its current form does not clearly depict the full workflow and should be redesigned for greater clarity.

---

> ### Author Rebuttal · Authors · 2025-07-30
>
> Thank you for your positive feedback and for recognizing the significance and potential of our method.
>
> > Q1: Clarity and presentation – Several aspects of the manuscript hinder readability.
>
> **A1:** We thank the reviewer for the helpful suggestions. For a and b, we will include key variable definitions and move Algorithm 1 to the main text in the revised version. For c, we will revise Figure 1 in the revised version to better reflect the complete UMPL workflow. The updated figure will explicitly show the teacher–student interaction, the three-stage training loop (pseudo-label generation, student update, teacher update), and the roles of Progressive Uncertainty Filtering (PUF) and Supervised Anchoring (SA), thereby enhancing the clarity and interpretability of the framework.
>
> >Q2: Could you elaborate on the computational resources your method requires?
>
> **A2:** Thanks for the question. All experiments were conducted on a single NVIDIA RTX 4090 GPU. We report results on AeroGTO model across three representative tasks—2D steady-state, 2D transient, and 3D simulation—as summarized below.
>
> Dataset|Darcy (bs=4)||NSM2d (bs=4)||Ahmed (bs=1)||
> -|-|-|-|-|-|-
> Method|Memory(M)|Training Time(h)|Memory(M)|Training Time(h)|Memory(M)|Training Time(h)
> Supervised|2297|0.45|9727|3.75|9375|3.53
> Pseudo Label|3843|1.08|16567|9.04|15947|8.73
> Mean Teacher|3843|1.21|16567|10.17|15947|9.57
> Noisy TS|2304|1.77|9813|13.21|9477|12.48
> Supervised+uq|3243|0.96|13578|7.69|12986|7.36
> UMPL|3257|2.21|13792|16.48|13290|15.56
>
> Here, Supervised+uq refers to the student model used in UMPL. As shown, **UMPL introduces almost no additional memory overhead**, thanks to the REINFORCE-style update that avoids the costly backpropagation chain from the student to the teacher. This significantly reduces computational and memory complexity, **at the cost of increased training time**.
>
> >Q3: Within the teacher–student framework, is there a risk of model collapse? How to protect against this?
>
> **A3:** We thank the reviewer for the insightful question. We agree that model collapse is a potential risk in teacher–student frameworks. To mitigate this, our design incorporates several mechanisms:
>
> - **Divergent Initialization Strategy:** The teacher and student models are initialized with different parameters, and during each training loop, they are exposed to independently sampled labeled data. This helps reduce the risk of early-stage over-alignment due to shared errors.
>
> - **Uncertainty-Guided Feedback Loop:** The student provides point-wise feedback based on aleatoric uncertainty, which captures spatial error sensitivity and directs the teacher’s updates toward high-error regions. This breaks blind imitation and reduces the chance of reinforcing incorrect signals.
>
> - **Uncertainty-Aware Pseudo Labels:** Pseudo labels are perturbed using the teacher’s epistemic uncertainty, replacing hard targets with uncertainty-aware guidance. This alleviates label bias and stabilizes convergence.
>
> - **Progressive Uncertainty Filtering (PUF):** Before being used for training, the teacher’s pseudo labels are filtered by uncertainty. Low-confidence samples are discarded, improving pseudo-label quality and enhancing training stability.
>
> We will include a detailed discussion of these safeguards in the revised version to better address this potential risk.

---

> > ### Comment · Reviewer_fwWx · 2025-08-06
> >
> > Thanks for answering my questions and providing a detailed response. I will maintain my score.

---

> > > ### Author Response · Authors · 2025-08-06
> > > **Appreciation for the constructive comments**
> > >
> > > We truly appreciate your valuable feedback, which has helped us strengthen our work. Thank you for your time and effort!

---

> ### Author Response · Authors · 2025-08-06
> **Looking forward to your reply**
>
> Dear Reviewer fwWx,
>
> We have provided new discussions according to your valuable suggestions, which have been absorbed into the revised manuscript. We hope that the new manuscript is made to be stronger with your suggestions.
>
> As the rebuttal deadline is approaching, we sincerely look forward to your reply. Thanks so much for your time and effort!
>
> Best Regards,
>
> Authors of Submission 10749

---

### Comment · Area_Chair_aA88 · 2025-08-03
**Author-Reviewer Discussion**

Dear Reviewers,

The author-reviewer discussion period is now open and will continue until August 6. Please review the authors’ rebuttal to determine whether it adequately addresses your concerns.  If you have further questions or comments, engage with the authors by acknowledging that you’ve read their response and providing additional feedback as needed

Sincerely,

Your AC

---

### Author Response · Authors · 2025-08-09
**Summary of the rebuttal and the major changes of revised manuscript**

Dear Reviewers, ACs, SACs, and PCs,

We would like to express our heartfelt gratitude for your invaluable time and meticulous attention in reviewing our manuscript. We appreciate that all four reviewers gave us positive feedback with insightful suggestions that can help us improve the quality and rigor of our work.

We have provided detailed responses to each reviewer’s concerns:

- **Reviewer fwWx** raised concerns on computational cost, clarity of presentation, and risk of model collapse in the teacher–student framework.

- **Reviewer VXJf** questioned the justification of key design choices (loss, embedding, MC Dropout, REINFORCE), clarity of figures, missing computational cost details, and the lack of comparison with other uncertainty estimation methods and statistical significance.

- **Reviewer 9hkT** raised concerns about MC-Dropout under non-Gaussian uncertainties, lack of calibration validation, overly clean benchmarks, and missing resource reporting.

- **Reviewer FbuM** focused on unclear SSL vs. uncertainty contribution, limited benchmark validation, missing overhead analysis, and insufficient robustness discussion.

For each, we have diligently addressed all concerns by conducting extensive additional experiments to support our responses. The main manuscript has been revised with improved descriptions and updates to figures. A detailed list of all changes is provided in the rebuttal, and additional experimental results are included in the appendix due to space constraints. Below, we summarize the key modifications made in both the rebuttal and the revised manuscript:

1. **Extended baselines:** Compared with seven model-agnostic uncertainty methods, our approach showed superior accuracy and uncertainty quality on unseen data.

2. **Extended experiments:** Replacing teacher–student modules with alternative uncertainty methods improved performance at higher cost, demonstrating the framework’s flexibility, while module refinement maintained strong results on non-Gaussian data.

3. **Further analyses:** Sensitivity tests on aleatoric quality, noise robustness, PUF threshold, and uncertainty injection, plus efficiency evaluation showing no extra GPU memory use but added feedback time.

4. **Manuscript revisions:** Updated Figures 1–2, clarified the core objective, refined formulations, added explanations of REINFORCE and encoder embedding, reported significance in the appendix, and expanded analysis on model collapse.

Overall, UMPL introduces (1) the first integration of pseudo-labeling with high-dimensional PDE modeling on non-uniform meshes for improved OOD generalization, (2) a role-aware teacher–student framework for robust and interpretable dual uncertainty estimation, and (3) an uncertainty-informed feedback mechanism that iteratively enhances pseudo-label quality in high-uncertainty regimes.

We kindly request that you carefully consider the contributions of our paper.

Best Regards,

Authors of Submission 10749

---

### Note · Authors · 2025-08-15

Dear Reviewers, ACs, SACs and PCs,

We thank all four reviewers for their valuable feedback that greatly helped us strengthen UMPL.

First of all, we appreciate that the reviewers acknowledge the advantages of our work: Innovative architecture-agnostic framework delivering robust performance across diverse settings (reviewer fwWx); Well-written paper with clear motivation and problem formulation (reviewer VXJf); Original teacher-student loop, mesh-aware pseudo-labels, and extensive experiments (reviewer 9hkT); Dual-uncertainty meta-learning with efficient, well-founded estimation and strong results (reviewer FbuM).

Secondly, we addressed reviewers’ concerns through targeted experiments, clarifications, and manuscript revisions:

1. **Extended baselines:** Benchmarked against seven model-agnostic uncertainty methods, achieving superior accuracy and uncertainty quality on unseen data.

2. **Additional experiments:** Tested alternative uncertainty modules (higher cost but improved performance) and refined existing ones for strong results on non-Gaussian data, highlighting framework flexibility.

3. **Further analyses:** Conducted sensitivity studies on aleatoric quality, noise robustness, PUF threshold, and uncertainty injection, alongside efficiency tests showing unchanged GPU memory usage with minimal feedback overhead.

4. **Manuscript updates:** Revised Figures 1-2, clarified objectives and formulations, added explanations (REINFORCE, encoder embedding), reported significance in the appendix, and expanded model-collapse analysis.

Thirdly, building on the rebuttal and discussions, we highlight UMPL’s key contributions: (1) Novel integration: The first to combine pseudo-labeling with high-dimensional PDE modeling on non-uniform meshes to boost OOD generalization; (2) Dual-uncertainty design: Employs a role-aware framework for robust and interpretable estimation; (3) Iterative feedback: Uses uncertainty-guided refinement to enhance pseudo-label quality in challenging regimes.

We sincerely thank all reviewers again for their constructive input, which enabled us to substantially improve the quality, clarity, and breadth of our work.

Yours sincerely,

Authors of Submission 10749

---

### Decision · Program_Chairs · 2025-09-17

**Decision:**

Accept (poster)

**Comment:**

This paper introduces an uncertainty-informed method for constructing robust surrogate models of PDE-governed physical systems under limited labeled data. Specifically, the authors propose a student–teacher model with an uncertainty-informed feedback loop designed to improve both uncertainty estimates and the quality of pseudo-labels. Extensive evaluations across seven tasks demonstrate that their method consistently outperforms the best existing semi-supervised regression approaches, even with limited labeled data.  The paper received four reviews: two accepts and two borderline accepts. All reviewers recognize the novelty of using an uncertainty-informed feedback mechanism in a teacher–student framework to simultaneously refine pseudo-labels and uncertainty estimates. They also highlight the strong empirical results demonstrated across multiple simulated and real-world datasets using different architectures.  Nonetheless, reviewers raise several concerns. These include the choice of UQ method for the teacher model, the computational complexity of the proposed approach, performance under noisy real-world data, questions about the statistical significance of the reported results, and the adequacy of the uncertainty evaluation.  The authors’ rebuttal is extensive and detailed, providing new experimental results that largely address these concerns.